# FIXED AGGREGATION FEATURES CAN RIVAL GNNS

## ABSTRACT

Graph neural networks (GNNs) are widely believed to excel at node representation learning through trainable neighborhood aggregations. We challenge this view by introducing Fixed Aggregation Features (FAFs), a training-free approach that transforms graph learning tasks into tabular problems. This simple shift enables the use of well-established tabular methods, offering strong interpretability and the flexibility to deploy diverse classifiers. Across 14 benchmarks, well-tuned multilayer perceptrons trained on FAFs rival or outperform state-of-the-art GNNs and graph transformers on 12 tasks – often using only mean aggregation. The only exceptions are the Roman Empire and Minesweeper datasets, which typically require unusually deep GNNs. To explain the theoretical possibility of non-trainable aggregations, we connect our findings to Kolmogorov–Arnold representations and discuss when mean aggregation can be sufficient. In conclusion, our results call for (i) richer benchmarks benefiting from learning diverse neighborhood aggregations, (ii) strong tabular baselines as standard, and (iii) employing and advancing tabular models for graph data to gain new insights into related tasks.

## 1 INTRODUCTION

Graph neural networks (GNNs) have become the standard approach for learning from graph based data, and in particular, for solving node classification. Most models follow the message-passing paradigm (Gilmer et al., 2017), where each node updates its representation by alternating neighborhood aggregation with learned linear combinations across multiple hops. This framework has been remarkably successful at combining node features with graph structure, driving applications in domains ranging from social networks to biology (Bongini et al., 2023; Sharma et al., 2024). Yet, it comes at the cost of high model complexity that poses challenges for interpretation. We ask the question whether this high complexity is really necessary.

Recent evidence (Luo et al., 2024; 2025a) shows that classic models, such as GCN (Kipf & Welling, 2017), GATv2 (Brody et al., 2022), and GraphSAGE (Hamilton et al., 2017), remain surprisingly competitive when equipped with proper hyperparameter tuning and standard optimization techniques. When carefully tuned, they can rival more sophisticated approaches, including state-of-the-art Graph Transformers (Wu et al., 2023; Deng et al., 2024; Kong et al., 2023; Wu et al., 2022; Chen et al., 2023; Rampášek et al., 2022; Shirzad et al., 2023) and models designed for heterophily (Zhu et al., 2020; 2021; Chien et al., 2021; Maurya et al., 2022; Li et al., 2022).

These results invite a closer look at which components of graph learning architectures are essential for strong performance, and thus raise a natural next question: *How relevant is learning the aggregation?* In fact, the field has invested heavily in learning increasingly complex convolution layers and attention mechanisms. In this paper we challenge that premise from first principles. Leveraging the Kolmogorov–Arnold representation theorem (Kolmogorov, 1957; Schmidt-Hieber, 2021), we give an explicit, lossless construction of neighborhood aggregations. Consequently, one can in theory encode neighbor features without discarding information. However, the same construction exposes a crucial gap between expressiveness and learnability: these lossless encoders are numerically brittle (e.g., sensitive to floating-point noise) and tend to produce "rough" embeddings that are ill-suited for standard classifiers on Euclidean space such as MLPs.

Surprisingly, we find that, standard, untrained aggregation operators—sum, mean, max, min—though not information-preserving, yield useful features without any learnable parameters. Building on this observation, we propose Fixed Aggregation Features (FAF) (§ 3): a training-free

aggregation pipeline that applies fixed aggregation functions—also referred to as "reducers"—over neighborhoods at multiple hops, concatenates the results into a tabular feature matrix, and then trains only a downstream classifier (e.g., an MLP). This data transformation brings several advantages: high interpretability (feature importance and ablations over hops/reducers), compatibility with the rich toolbox of tabular learning (designed to handle noise, class imbalance, feature selection, etc.), architectural flexibility, and reduced training compute.

Empirically, FAFs combined with well-tuned MLP classifiers are competitive on 12/14 common node-classification benchmarks, including citation networks (McCallum et al., 2000; Sen et al., 2008; Namata et al., 2012), coauthor and Amazon co-purchase graphs (Shchur et al., 2018), Wikipedia (Mernyei & Cangea, 2020; Rozemberczki et al., 2021), and other heterophilous datasets (Platonov et al., 2023b). Performance truly trails only on Minesweeper and Roman-Empire, where the best GNNs rely on linear residual connections; in fact, the remaining gap aligns with the gains from residuals reported by Luo et al. (2024). This pattern suggests that these datasets (Minesweeper and Roman-Empire) benefit from hop-specific aggregations. While these GNNs profit from many layers (10–15), the best-performing FAFs use only 2–4 hops. But why do FAFs work so well in the other cases? Our theoretical analysis of the employed aggregation functions (§ 4) and our empirical findings (§ 5) suggest that, for most benchmarks, the relevant signal is concentrated within hops 0–2; on hops 0–1, sum and mean preserve information. At higher hops, different reducers are complementary, but the information gain from min/max diminishes.

FAFs also let us examine datasets from an optimization-first viewpoint without hard-to-interpret architectural factors (§ 3.1). By converting neighborhoods into tabular features, we decouple representation from optimization and enable standard interpretability tools (e.g., feature importance (Lundberg & Lee, 2017)) to identify which hop distances and reducers carry signal. Beyond revisiting the homophily–heterophily dichotomy or one-hop informativeness (Platonov et al., 2023a; Zheng et al., 2024; 2025), our method supports a richer characterization of interaction patterns—how signal varies across scales, which effects are additive vs. redundant, and where long-range dependencies matter. The tabular view also makes it natural to augment features with network-science descriptors (Blöcker et al., 2025) and neighborhood-masking features inspired by graph rewiring (Rubio-Madrigal et al., 2025) or computational-graph splitting (Roth et al., 2025).

Together, our results suggest that many benchmarks do not require sophisticated learned aggregations, and that a large portion of GNN performance can be matched by powerful tabular baselines built from fixed, transparent summaries. FAFs can serve both as a strong baseline and as a diagnostic tool for graph benchmarks in this setting. In summary, here are our main contributions:

1. **Theory:** We construct lossless neighborhood aggregations via Kolmogorov–Arnold representations, clarifying that learnability and numeric stability—not just expressiveness—govern practical success. Moreover, we analyze what information common reducers extract from neighborhoods, revealing information preservation at 0 and 1 hops and diminishing information with higher depth for min/max.

2. **Method:** We introduce FAFs, which convert graph data into a tabular task by stacking fixed multi-hop aggregations, offering an interpretable framework to study the interplay between graph structure, features, and the task.

3. **Empirical evidence:** FAFs match or exceed classic GNNs on 12/14 standard benchmarks. Our experiments further corroborate our analysis, finding low hop features to be more important and diminishing information at higher depth.

4. **Implications:** Our findings question the necessity of learned neighborhood aggregation on current standard benchmarks, motivate strong tabular baselines for graph data, and open a path to more interpretable, efficient graph learning—and to designing harder benchmarks that genuinely benefit from aggregation learning.

## 2 RELATED WORK

**Simplifying GNNs.** A growing body of work shows that much of a GNN's power can be retained—even improved for some tasks—when message passing is simplified or fixed. Early evidence comes from Kipf & Welling (2017), inspiring lines of work where aggregation layers are

frozen or randomized: Ramachandra et al. (2025) obtain competitive node classification on relational graphs by aggregating randomly transformed random features; Kelesis et al. (2025) analyze partially frozen GCNs, showing that fixing aggregation can mitigate over-smoothing and ease optimization; and GESN (Gallicchio & Micheli, 2010) compute node embeddings via a dynamical system with randomly initialized reservoir weights, after which only a linear readout is trained for node classification (Micheli & Tortorella, 2024). Another simplification comes from SGC (Wu et al., 2019), which remove nonlinearities from a GCN, yielding a model equivalent to applying a low-pass graph filter followed by a linear readout. For link prediction (Qarkaxhija et al., 2024), SGCNs are found to be better than GCNs, but even removing the classifier linear layer—and thus all trainable parameters—provides a good baseline. A fixed aggregation scheme is also used in APPNP (Gasteiger et al., 2019): an MLP is first trained to produce node embeddings, and a subsequent Personalized PageRank–based propagation is applied; although the propagation itself is fixed, it remains in the computation graph, so gradients flow during backpropagation. And for graph classification on non-attributed graphs, Cai & Wang (2019) show that first-neighborhood statistics with an SVM form a surprisingly strong baseline, but performance lags on attributed graphs. In contrast, we focus on node classification with rich node features. We aggregate across all hops and concatenate these as inputs, and we place a more powerful, and more carefully tuned classifier on top (an MLP), all of which are necessary for our results—as shown in our empirical results, such as in Tables 7 and 8. With it we highlight the value of concatenating dependent but informative hop-wise features (Reddy et al., 2025), and the benefits of overparameterization on graphs—echoing evidence from random-feature models (Donghi et al., 2024).

**Benchmarking GNNs.** Our work also connects to the growing literature on properly benchmarking GNNs and what constitutes a meaningful graph-learning dataset. For graph classification, Errica et al. (2020) show that, under controlled protocols, simple and even structure-agnostic baselines can rival complex GNNs, suggesting that common benchmarks often fail to exploit graph structure. Recent analyses likewise warn that graph learning risks losing relevance without application-grounded benchmarks (Bechler-Speicher et al., 2025) and principled criteria for dataset quality beyond accuracy (Coupette et al., 2025). On the dataset side, Bazhenov et al. (2025) recently introduced industrial node property prediction benchmarks with graph-agnostic baselines. Their neighborhood feature aggregation (NFA), which augments tabular models with one-hop aggregated neighbor statistics, can be seen as a one-hop instance of our FAF construction. Concurrently, Anonymous (2025) propose a suite of benchmarks to standardize graph-learning evaluation. In this context, our FAF approach serves as a simple stress test of whether proposed graph benchmarks genuinely benefit from learned message passing, and we argue that such well-tuned, fixed, multi-hop baselines should be routinely included when assessing new graph models and datasets.

**GNN aggregation functions.** Classical message-passing GNNs differ primarily in how they aggregate neighbor features under permutation invariance. Sum, mean, and max are the canonical choices, with injectivity and stability trade-offs of each of them tied to their multiset representations (Xu et al., 2019). Beyond single operators, principal neighborhood aggregation (PNA) mixes several base reducers with degree-aware scalers to boost expressiveness and well-conditioning for continuous reducers (Corso et al., 2020). Attention mechanisms instantiate learned weighted sums (Veličković et al., 2018; Brody et al., 2022), although it has been shown that they suffer from trainability problems, including small relative gradients on the attention parameters, slowed-down layer-wise training speed, and the inability to mute neighbors (Mustafa et al., 2023; Mustafa & Burkholz, 2024a;b). Our perspective is complementary: We study fixed reducers whose strength comes from (i) their information preservation and (ii) their separability by a powerful downstream classifier. This decoupling clarifies what must be learned (the readout) versus what can be fixed (the propagation), and it aligns with our empirical finding that stronger, well-tuned classifiers capitalize on rich, concatenated neighborhood views. Because of this, we argue that good optimization and learnability is as important as expressivity results. In line with this argument, Gorishniy et al. (2022) argue for tabular data that having the right embeddings for continuous features is key to closing the gap between transformer-like architectures and feed forward networks, proposing a lossless piecewise linear embedding to improve the trainability of the latter.

**Kolmogorov-Arnold theorem.** The Kolmogorov–Arnold representation (Kolmogorov, 1957) admits several equivalent formulations that reduce multivariate functions to compositions of univariate

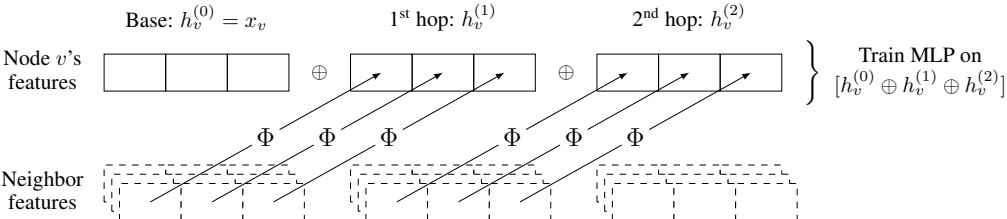

Figure 1: Fixed Aggregation Features (FAFs) are calculated as a pre-processing step, concatenated to the input ($\oplus$), and fed to an MLP. If the aggregation function $\Phi$ is injective, the neighborhood information is preserved. The Kolmogorov-Arnold representation theorem ensures the existence of such a function, although simple reducers are empirically more amenable for optimization.

functions. Recent architectures explicitly instantiate such decompositions with learnable spline-based univariate components and linear mixing (Liu et al., 2025; Carlo et al., 2025). In contrast, we use a specific fixed-aggregation formulation with predetermined aggregation weights and a fixed univariate encoding (Schmidt-Hieber, 2021), so that any multivariate function $f$ can be learned exclusively from a single univariate readout $g$ applied to a fixed weighted sum of univariate transforms.

## 3 FIXED AGGREGATION FEATURES

Node neighborhoods can be compressed into single-node features, eliminating the need to learn feature embeddings before every layer of message passing. Our approach, Fixed Aggregation Features (FAFs), recursively constructs and concatenate features via a set of reducers $\mathcal{R} \subset \{\text{mean}, \text{sum}, \text{max}, \text{min}, \text{std}, \ldots\}$ in the following way:

$$h_v^{(0,r)} = x_v, \qquad h_v^{(k,r)} = r\Big(\big\{ h_u^{(k-1,r)} : u \in N(v) \big\}\Big), \tag{1}$$

where $k \in \{1, \ldots, K\}$ and $r \in \mathcal{R}$. We then train an MLP on the concatenated representation

$$z_v = x_v \oplus \left( \bigoplus\nolimits_{r \in \mathcal{R}} \bigoplus\nolimits_{k \in \{1,\ldots,K\}} \left( h_v^{(k,r)} \right) \right) \tag{2}$$

with input dimensionality $|x_v| \cdot (1 + |\mathcal{R}| \cdot K)$ per node $v$. Figure 1 illustrates the case $\mathcal{R} = \{\Phi\}$ with $K = 2$. If the reducers are injective, then the neighborhood information at each depth is preserved in $z_v$. This waives the need for aggregating learned embeddings in GNNs, thus transforming graph data into high-dimensional tabular data. Our analysis explains why this is theoretically possible (§ 4). Additionally, in our experiments (§ 5), we show that MLPs trained on FAFs can match the performance of classic GNNs on most standard node-classification benchmarks and, by comparing with Luo et al. (2024), of Graph Transformers and heterophily-aware models.

### 3.1 ADVANTAGES OF TABULAR OVER GRAPH DATA REPRESENTATION

We now turn to further benefits of the tabular view: interpretability, optimization, efficiency, and augmentations.

**Interpretability.** Our construction concatenates each node's original features with $K$-hop neighborhood statistics and feeds this expanded representation to a tabular classifier. This setup surfaces feature and hop aggregation factors explicitly, enabling us to assess their contributions using the widely-used toolbox for tabular interpretability. For instance, we can analyze effects across hops by examining feature importance of the MLP. As an illustration, we compute Shapley

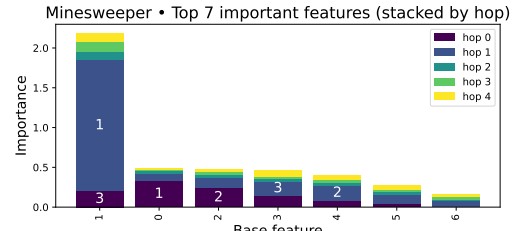

Figure 2: SHAP feature importance for Minesweeper, stacked by hop. Numbers on the stacked bars indicate the ranking of that particular feature on that particular hop.

Additive Explanations (SHAP) (Lundberg & Lee, 2017) on Minesweeper with mean aggregation (Fig. 2), one of the two datasets where FAFs lag. In this dataset, labels are bombs, feature 0 masks other features, and features 1–6 one-hot encode the number of neighboring bombs. The top signal is the hop-1 mean of feature 1, i.e., the fraction of neighbors whose local bomb count is null. When this proportion is greater than zero, the model knows that the node cannot have a bomb; when it is zero, all its neighbors observe bombs so there is a possibility of having a bomb. This heuristic does not completely solve the problem, as neighbors can be merely observing second-hop bombs, creating ambiguity that likely underlies residual errors. The model also correctly gives importance to the number of masked neighbors (hop-0 feature 0). These attributions clarify where the model succeeds and where it fails. For comparison, we report SHAP importances for two other datasets in Fig. 5 in the appendix: Pubmed (homophilic) and Amazon-Ratings (heterophilic).

Rather than explaining a particular classifier, one can aim to explain the dataset—localizing which hops and features carry signal independent of model choice. Following Donnelly et al. (2023), noisy tabular datasets often admit a "Rashomon set" of comparably well-performing models. Accordingly, feature importance is better assessed over this set—preferably constrained to simpler or sparser models—than from a single fit. This lens may offer a principled way to interrogate graph data beyond feature homophily–heterophily (Zheng et al., 2024) or strictly one-hop neighborhoods (Zheng et al., 2025), though its current implementation requires binarized data on regression tasks.

**Optimization.** GNNs usually exhibit early overfitting, where training accuracy converges almost immediately while validation and test accuracy plateau or even decay. Thus, the best validation accuracy is often achieved before relevant aggregations are learned. This might partially explain why FAFs can often compete with trained GNNs: They avoid overfitting aggregations. GNNs can also suffer from ineffective aggregation learning (Mustafa et al., 2023; Mustafa & Burkholz, 2024b), so their potential to outcompete FAFs is likely underexplored due to trainability issues. By contrast, optimization on tabular data (like FAFs) is better tractable and understood by standard toolkits.

**Efficiency.** Precomputing aggregation once and then training an MLP on top is far more scalable than repeatedly running message-passing layers and backpropagating through them, as required in GNNs. However, as the number of reducers, original features, and hops in FAF increases, so does the input dimensionality, which in turn enlarges the parameter count of the MLP's first layer. This issue could be mitigated through common feature reduction techniques. For the original features, we report the average training runtimes of our FAF and GNN models in Table 5. FAFs are generally more efficient, particularly when using a single reducer.

**Augmentations.** Adding more features does not improve accuracy monotonically. Beyond a point, some feature selection is needed. Still, the tabular view lets us concatenate diverse features atop the aggregations. In addition to concatenating multiple reducers and hops, one can append structural statistics such as degree, centrality, and other network-science metrics (Blöcker et al., 2025). Our framework is also compatible with pre-processing graph rewiring, where aggregation is computed on a modified adjacency matrix e.g. to fight over-squashing (Topping et al., 2022; Jamadandi et al., 2024). But unlike standard rewiring, we can concatenate the rewired features instead of replacing the originals. As fixed aggregations can suffer from similar issues as trainable GNNs, FAFs can also benefit from proposed remedies. To extract more precise information from complex environments, we examine a feature similarity-based rewiring loosely based on Rubio-Madrigal et al. (2025), where edges of negative feature cosine similarity between nodes are dropped. We then append features aggregated on the rewired graph, or split edges into positive/negative sets and aggregate separately, inspired by computational-graph splitting that helps fight over-smoothing (Roth et al., 2025). Results (Table 10) show that on datasets already helped by mean aggregation, adding these features yields larger gains than substituting them; e.g., WikiCS surpasses classic GNNs where other FAF variants do not. These augmentations not only improve performance but also help disentangle where the gains come from: additional extracted signal versus changes to the optimization of graph models. We therefore advocate FAFs as baselines for methods that modify the aggregation component of GNNs—akin to analyses of SGC and GESN (Micheli & Tortorella, 2025), though in our case we obtain benefits from these operations.

## 4 THEORETICAL FOUNDATIONS: DOES AGGREGATION NEED LEARNING?

Let $G = (V, E)$ be a graph with node features $\mathcal{X} \in \mathbb{R}^F$. A neighborhood function is a map $f : \mathcal{M}(\mathbb{R}^F) \to \mathbb{R}$ acting on the multiset $X_v := \{\mathcal{X}_u : u \in N(v)\}$ for $v \in V$. We seek a fixed encoder $\Phi$ such that we can learn any neighborhood function via a univariate map $g$ with $f = g \circ \Phi^{-1}$. This enables tabular classifiers to learn graph data losslessly.

### 4.1 WHAT INFORMATION IS PRESERVED BY STANDARD AGGREGATIONS?

FAFs apply a transformation of neighborhoods that is not learnable, which raises the question of what information gets lost by the aggregation. Permutation invariant aggregations treat graph neighborhoods as multisets consisting of feature vectors of neighbors. Accordingly, they extract distribution information and forget about the identity of specific neighbors. This property is usually regarded as helpful inductive bias and therefore of no concern. Our next theorems analyze which information is preserved by sum and mean aggregations from these multisets. To do so, we first generalize Lemma 5 by Xu et al. (2019) for one-hot encoded discrete features to orthogonal features. Combined with the fact that hop features are concatenated, this insights establishes that information from the 1-hop neighborhood can be preserved.

**Theorem 1** (1-hop aggregation). *Assume the features $\mathcal{X}$ are orthogonal. Then, the function $h(X) = \sum_{x \in X} x$ defined on multisets $X \subseteq \mathcal{X}$ of bounded size is injective. Moreover, any multiset function $f$ can be decomposed as $f(X) = g\left(\sum_{x \in X} x\right)$ for some function $g$.*

The proof is given in the appendix A.2. Note that a multiset $X \subseteq \mathcal{X}$ is characterized by the count $n_x$ of elements that have specific features $x$. These counts can also be extracted from the sum $h(X)$ (as demonstrated in the proof). Consequently, any multiset function $f$ transforms such counts by $f(n_x)$. The function $g$ would thus first extract the counts from the sum $h(X)$ and then apply $f$ to the counts. If the features of a node $v$ include its degree $d_v$, then mean aggregation contains the same information, as a classifier can learn to multiply $h(X) = 1/d_v \sum_{x \in X} x$ by $d_v$. In contrast, max and min aggregations extract whether at least one neighbor has a specific feature property. They focus on the tails of distributions rather than full neighborhoods.

**Information loss for $k$-hops.** One might hope that the above theorem also applies to aggregations from hop $k$ to $k + 1$. The orthogonality assumption, however, is essential and no longer met by the aggregated neighbor features $h_n^{(k)}$ for $k \geq 1$. As a consequence, from $k \geq 2$, not all information about the distribution of features across neighbors is preserved, as Figure 4 exemplifies. In particular, $h_1^{(2)}$ captures neither the degrees of its neighbors, nor the number and types of second-hop neighbors associated with each first-hop neighbor. Even so, aggregation still extracts useful information, and different aggregations concatenate complementary properties of neighborhoods.

- **Sum aggregation:** Sums count, for each of the $n$ distinct orthogonal feature vectors $x_f$, how many nodes in the $k$-hop neighborhood exhibit feature $f$. A classifier can extract it by computing $x_f^T h_v^{(k)}$. Note that nodes reachable through multiple length-$k$ paths are counted multiple times.

- **Mean aggregation:** Means can partially distinguish neighbors with different degrees by considering the fraction of nodes that exhibit a specific feature vector $x_f$. The quantity $x_f^T h_v^{(k)}$ weights each node $i$ with feature $x_f$ by $1/d_i$. Note that nodes reachable through multiple length-$k$ paths are again counted with multiplicity.

- **Max aggregation:** Max aggregation on one-hot encoded features returns whether at least one node within $k$ hops has a given feature. For large neighborhoods as $k$ increases, this indicator quickly saturates, so increasing hops adds little further information. The same reasoning applies when taking the maximum entry of the orthogonal features.

- **Min aggregation:** The same reasoning applies to the minimum as to the maximum: It indicates whether any node within $k$ hops lacks the feature, and increasing $k$ adds little further information.

### 4.2 LOSSLESS NEIGHBORHOOD AGGREGATION

When node features are real-valued in general, Corso et al. (2020) show that no single *continuous*, permutation-invariant aggregation function can be lossless for all multiset functions. This mirrors

a classical topological obstruction due to Netto (1879): There is no continuous bijection $\mathbb{R} \to \mathbb{R}^2$ (Dauben, 1975). However, there can exist discontinuous bijections, namely space filling functions. We adopt a concrete construction based on ternary expansions and the Cantor set, adapted from a Kolmogorov-Arnold representation variant from Theorem 2 by Schmidt-Hieber (2021).

**Theorem 2** (Kolmogorov-Arnold representation from Thm. 2 of Schmidt-Hieber (2021)). *For any fixed $d \geq 2$, there exists a monotone function $\phi : [0, 1] \to \mathcal{C}$ (the Cantor set) such that the map $\Phi(x_1, \ldots, x_d) = 3 \sum_{p=1}^{d} 3^{-p} \phi(x_p)$ is injective on $[0, 1]^d$. Moreover, for every continuous $f : [0, 1]^d \to \mathbb{R}$ there exists a continuous $g : \Phi([0, 1]^d) \to \mathbb{R}$ with $f(x_1, \ldots, x_d) = g(\Phi(x_1, \ldots, x_d))$.*

Theorem 2 isolates all required discontinuity into a *fixed* aggregation. While $\Phi$ is not continuous, its inverse is, which makes the learnable part $g := f \circ \Phi^{-1}$ inherit the continuity properties of $f$. Schmidt-Hieber (2021) has also quantified how much information is lost if $g$ is learned instead of $f$. For $f$ $\beta$-smooth with $\beta \leq 1$, there is no difference in the rate of approximation. However, for higher order smoothness, the multivariate and univariate function approximation may vary. Note that this aggregation even remembers node identities. From this theorem we can learn the following insight:

> A lossless, fixed, even univariate neighborhood aggregation function exists, but it has to be discontinuous for general continuous features.

### 4.3 IMPLICATIONS AND OPEN CHALLENGES

When we encode neighborhoods via the injective function $\Phi$ and learn $g$ so that $f = g \circ \Phi^{-1}$, the information content, smoothness properties, and approximation rates of the neighborhood function $f$ transfer to $g$. However, this theoretical sufficiency does not guarantee strong empirical performance when $\Phi$ is used directly as a reducer for FAF (see Table 9). In Appendix A.1, we visualize how $\Phi$ maps 2D circles into the univariate Cantor set, and how $\Phi^{-1}$ can recover them continuously. We also compare against mean and std, and observe that $\Phi$ pushes inputs that are close together into far-apart representations, whereas mean and std bring together far-apart inputs that share commonalities. It is the case that, in practice, the simple statistics studied in § 4.1 often provide distributional summaries that downstream classifiers exploit more effectively.

An ideal aggregation function would be both injective, like $\Phi$, and would extract useful statistical insights, like mean. It is still an open challenge to design, or potentially learn, efficient embeddings that extract relevant information from graph neighborhoods, while easing the learning problem for the classifier (Burkholz et al., 2022). One might expect GNNs to learn such representations end-to-end without overfitting. Our experiments with FAFs (Tables 2, 6) suggest—despite some information loss at iterative hops—that simple reducers suffice for most standard node-classification benchmarks.

Experimentally, we find that mean aggregation alone is often among the top performers. This suggests that neighborhood feature distributions provide most task-relevant signal, and that neighbor degrees encode useful structural information, helping to distinguish the contribution of distinct neighbors. Consistent with this, the most relevant information is already provided in the immediate neighborhood ($k = 0, 1$, see Table 6) and the concatenation of this information is key so that it is not lost by repeated aggregations (see Table 8). Consequently, information loss at larger $k$ is of little practical concern—except for two datasets that appear to require subtler information. Taken together, these observations motivate the following hypothesis.

> Hypothesis: For most standard node-classification benchmarks, either the predictive signal is already concentrated within the first one or two hops, or current GNNs struggle to learn layer-wise aggregations that extract relevant information beyond mean or sum.

The first part of this hypothesis underscores the need for more real-world datasets where long-range interactions and richer aggregations matter, supporting prior calls to revisit benchmark design (Errica et al., 2020; Bechler-Speicher et al., 2025). Although some tasks (like Roman Empire) benefit from long-range signal (Topping et al., 2022), making deep graph models work reliably remains a challenge. Recent evidence indicates that graph models generally struggle to capture interactions beyond roughly 13 hops, irrespective of over-smoothing, over-squashing, or vanishing gradients (Zhou et al., 2025).

Table 1: Test accuracy on node classification: Best validation FAF against classic GNNs.

| Dataset | computer | photo | ratings | chameleon | citeseer | coauthor-cs | coauthor-physics |
|---|---|---|---|---|---|---|---|
| GCN | 93.58 ± 0.44 | 95.77 ± 0.27 | 53.86 ± 0.48 | **44.62 ± 4.50** | **72.72 ± 0.45** | 95.73 ± 0.15 | **97.47 ± 0.08** |
| GAT | 93.91 ± 0.22 | 96.45 ± 0.37 | **55.51 ± 0.55** | 42.90 ± 5.47 | 71.82 ± 0.65 | 96.14 ± 0.08 | 97.12 ± 0.13 |
| SAGE | 93.31 ± 0.17 | 96.17 ± 0.44 | 55.26 ± 0.27 | 43.11 ± 4.73 | 71.82 ± 0.81 | **96.21 ± 0.10** | 97.10 ± 0.09 |
| MLP | 87.75 ± 0.42 | 93.62 ± 0.36 | 49.04 ± 0.39 | 38.59 ± 3.29 | 57.22 ± 2.25 | 93.80 ± 0.19 | 96.02 ± 0.16 |
| FAF$_{\text{bestval}}$ | **94.01 ± 0.21** | **96.54 ± 0.13** | 55.09 ± 0.24 | 42.96 ± 2.45 | 70.48 ± 1.24 | 95.37 ± 0.17 | 97.05 ± 0.18 |

| Dataset | cora | minesweeper | pubmed | questions | roman-empire | squirrel | wikics |
|---|---|---|---|---|---|---|---|
| GCN | **84.38 ± 0.81** | 97.48 ± 0.06 | 80.00 ± 0.77 | 78.44 ± 0.23 | **91.05 ± 0.15** | 44.26 ± 1.22 | 80.06 ± 0.81 |
| GAT | 83.02 ± 1.21 | 97.00 ± 1.02 | 79.80 ± 0.94 | 77.72 ± 0.71 | 90.38 ± 0.49 | 39.31 ± 2.42 | **81.01 ± 0.23** |
| SAGE | 83.18 ± 0.93 | **97.72 ± 0.70** | 77.42 ± 0.40 | 76.75 ± 1.07 | 90.41 ± 0.10 | 40.22 ± 1.47 | 80.57 ± 0.42 |
| MLP | 58.56 ± 1.75 | 51.74 ± 0.83 | 68.22 ± 0.96 | 70.40 ± 1.17 | 66.43 ± 0.12 | 39.11 ± 1.93 | 72.98 ± 0.49 |
| FAF$_{\text{bestval}}$ | 82.84 ± 0.63 | 90.00 ± 0.39 | **80.96 ± 1.06** | **78.69 ± 0.50** | 78.11 ± 0.38 | **44.59 ± 1.62** | 80.25 ± 0.34 |

The second part of the hypothesis concerns the ability of GNNs to actually realize useful aggregations in practice. For instance, GNNs may not move far enough from their initializations. Indeed, the two datasets on which GNNs hold an advantage require linear residual transformations to realize that gap (Luo et al., 2024). Prior work also shows that GATs cannot flexibly adjust attention to shut off unhelpful neighbors (Mustafa & Burkholz, 2024b). This supports our results on rewiring the adjacency matrix before aggregation (Table 10). If GATs could learn to prune the edges that we manually drop, they would enjoy similar gains.

We see opportunity for future work along three fronts that build directly on our findings:

1. Feature/reducer engineering: FAFs highlight untapped potential for designing meaningful node features that encode graph structure, require less learning, potentially preserve more—but ideally only relevant—information, and allow for higher learning efficiency. In combination with partial feature learning, they might form the basis of a new generation of graph based learning architectures.

2. Moving beyond injectivity: As our theory and empirical results highlight, improving GNN expressiveness and thus injectivity alone is not likely to inspire practical improvements on current benchmarks, as those can be competitively solved even with simple, non-injective aggregation. We therefore call for a shift in focus from mere injectivity to other learning properties—a theoretical gap to be addressed not only for FAFs but for GNNs in general.

3. Benchmarks: Enough information to solve current benchmark tasks is already contained in early hops and can be extracted with simple, non-injective aggregation. If we really want to showcase the capabilities of GNNs to learn meaningful features, we need more difficult benchmarks that require this ability.

> In theory, fixed information-preserving aggregations can reduce graph learning to tabular prediction. In practice, task relevant representations and information preservation are a challenge. Progress likely requires both more amenable reducers and better tasks for evaluation.

## 5 EXPERIMENTS

### 5.1 COMPARISON TO CLASSIC GNNS

**Performance of FAFs.** Table 1 reports test performance for classic GNNs—GCN (Kipf & Welling, 2017), GATv2 (Brody et al., 2022), and GraphSAGE (Hamilton et al., 2017)—versus our approach, which feeds Fixed Aggregation Features (FAFs) into MLPs (Figure 1 and Eq. 1). We aggregate up to the same hop depth as the GNN baselines. As a control, we include an MLP baseline with zero-hop aggregation, which performs substantially worse than all other models. We obtain the best FAF variant from validation results, shown in Table 2. FAF$_4$ uses the reducers $\mathcal{R} = \{\text{mean, sum, max, min}\}$, and is tuned with exactly the hyperparameter grid from Luo et al. (2024); this makes our results directly comparable to their Graph Transformers and heterophily-aware architectures, where they find that classic GNNs can also rival them. Additional FAF variants include

Table 2: Validation accuracy on node classification: FAFs against classic GNNs.

| Dataset | computer | photo | ratings | chameleon | citeseer | coauthor-cs | coauthor-physics |
|---|---|---|---|---|---|---|---|
| GCN | 92.58 ± 0.10 | 95.42 ± 0.11 | 54.01 ± 0.23 | 48.15 ± 2.35 | **70.36 ± 0.09** | 95.32 ± 0.07 | **97.16 ± 0.07** |
| GAT | 92.86 ± 0.06 | 95.93 ± 0.15 | 55.56 ± 0.68 | 46.97 ± 2.07 | 69.52 ± 0.27 | 95.30 ± 0.08 | 97.11 ± 0.03 |
| SAGE | 92.33 ± 0.17 | 95.60 ± 0.16 | **55.90 ± 0.54** | 46.22 ± 2.12 | 68.48 ± 1.05 | **95.51 ± 0.04** | 97.02 ± 0.09 |
| MLP | 87.89 ± 0.13 | 93.33 ± 0.07 | 48.98 ± 0.72 | 41.43 ± 1.77 | 56.80 ± 1.24 | 93.70 ± 0.07 | 95.89 ± 0.02 |
| $FAF_4$ | 93.05 ± 0.04 | **96.34 ± 0.07** | 55.53 ± 0.43 | **48.51 ± 2.31** | 67.28 ± 0.64 | 94.93 ± 0.07 | 96.83 ± 0.01 |
| $FAF_{mean,std}$ | 93.04 ± 0.13 | 96.23 ± 0.08 | 55.11 ± 0.40 | 48.42 ± 1.64 | 67.20 ± 0.28 | 94.94 ± 0.07 | 96.84 ± 0.03 |
| $FAF_{mean}$ | **93.16 ± 0.04** | 96.06 ± 0.10 | 53.78 ± 0.52 | 47.99 ± 2.02 | 66.92 ± 0.87 | 95.20 ± 0.14 | 97.00 ± 0.04 |
| $FAF_{max,std}$ | 92.32 ± 0.08 | 95.80 ± 0.04 | 55.70 ± 0.45 | 48.42 ± 2.14 | 66.64 ± 0.54 | 95.04 ± 0.04 | 96.56 ± 0.03 |
| $FAF_{max}$ | 91.93 ± 0.04 | 95.60 ± 0.04 | 55.63 ± 0.29 | 48.06 ± 2.30 | 66.56 ± 0.50 | 95.19 ± 0.13 | 96.54 ± 0.01 |
| $FAF_{sum}$ | 90.95 ± 0.04 | 94.88 ± 0.04 | 53.48 ± 0.59 | 47.29 ± 1.92 | 67.84 ± 1.45 | 95.13 ± 0.09 | 96.65 ± 0.05 |
| $FAF_{std}$ | 92.50 ± 0.06 | 95.86 ± 0.10 | 55.31 ± 0.32 | 47.27 ± 2.15 | 63.44 ± 0.17 | 95.01 ± 0.12 | 96.76 ± 0.02 |

| Dataset | cora | minesweeper | pubmed | questions | roman-empire | squirrel | wikics |
|---|---|---|---|---|---|---|---|
| GCN | 81.28 ± 0.33 | 97.36 ± 0.46 | 79.08 ± 0.23 | 78.63 ± 0.23 | **91.14 ± 0.58** | 44.88 ± 1.27 | 81.52 ± 0.37 |
| GAT | 81.16 ± 0.52 | 97.08 ± 1.16 | 78.84 ± 0.52 | 78.12 ± 1.03 | 90.49 ± 0.68 | 43.30 ± 1.43 | **82.38 ± 0.57** |
| SAGE | 81.32 ± 0.41 | **97.68 ± 0.63** | 78.88 ± 0.91 | 77.35 ± 1.09 | 90.44 ± 0.66 | 40.58 ± 1.17 | 82.27 ± 0.38 |
| MLP | 62.68 ± 1.15 | 51.12 ± 0.93 | 71.12 ± 0.52 | 71.58 ± 1.46 | 66.28 ± 0.27 | 40.57 ± 0.92 | 74.86 ± 0.33 |
| $FAF_4$ | 82.84 ± 0.43 | 89.63 ± 1.03 | 79.08 ± 0.36 | **79.53 ± 1.12** | 78.68 ± 0.19 | 47.31 ± 1.39 | 81.92 ± 0.43 |
| $FAF_{mean,std}$ | **83.36 ± 0.17** | 89.18 ± 0.71 | **81.28 ± 0.30** | 77.32 ± 0.36 | 77.59 ± 0.41 | 47.30 ± 1.32 | 81.37 ± 0.51 |
| $FAF_{mean}$ | 83.28 ± 0.30 | 89.89 ± 0.93 | 81.16 ± 0.97 | 78.53 ± 0.87 | 76.67 ± 0.36 | 46.29 ± 1.50 | 81.58 ± 0.46 |
| $FAF_{max}$ | 81.80 ± 0.42 | 86.08 ± 0.77 | 77.48 ± 0.30 | 79.15 ± 0.86 | 75.06 ± 0.14 | 46.47 ± 1.38 | 80.30 ± 0.56 |
| $FAF_{max,std}$ | 82.08 ± 0.33 | 87.83 ± 0.63 | 78.28 ± 0.30 | 78.86 ± 0.89 | 76.19 ± 0.26 | **47.44 ± 1.51** | 80.46 ± 0.53 |
| $FAF_{sum}$ | 82.60 ± 0.65 | 89.86 ± 0.85 | 79.40 ± 0.57 | 78.12 ± 0.27 | 77.13 ± 0.23 | 46.85 ± 1.28 | 78.17 ± 0.23 |
| $FAF_{std}$ | 81.40 ± 0.51 | 88.20 ± 0.52 | 80.00 ± 0.40 | 76.25 ± 0.53 | 73.95 ± 0.49 | 45.91 ± 1.32 | 77.65 ± 0.31 |

mean+std, mean only, max+std, max only, sum only, and std only. The best overall result is shown in **bold**, the second best is underlined. More details on the setup are given in § B, as well as all test accuracy counterparts in Table 11 in § D. We also show all training, validation, and test curves of all datasets for $FAF_4$ and GCN in Fig. 6.

Overall, we improve on 5 datasets, match within error or 1% on another 5, and trail on 4. On most datasets, $FAF_4$ performs comparably to mean+std. Among the ones within 1%, we have Coauthor-CS and Coauthor-Physics (Figures 6f, 6g), which are are the largest and most feature-rich; targeted feature selection may close the gap. Among the 4 trailing datasets, two are homophilic and two heterophilic; the homophilic tasks are close to parity. Citeseer exhibits optimization instability (Fig. 6e), and Cora has a large test-validation gap in GCNs (Fig. 6h), not present in any other. The two heterophilic datasets, Minesweeper and Roman-Empire (Figures 6i, 6l) show larger performance drops. This behavior mirrors the decrease reported by Luo et al. (2024) when residual connections are removed. Notably, the best-performing FAFs on these two datasets use far fewer hops (4 and 2) than the GNN baselines (15 and 10), suggesting that key signal lies at longer ranges. The shallower FAFs under-aggregate relative to what those tasks require, but adding extra hops does not provide extra information, as discussed in § 4.1. We also show it in Table 6, where we concatenate up to different amount of hops. In fact, most datasets peak at $K = 2$, and either plateau or decrease in performance.

**Best hyperparameters.** All FAF variants benefit from normalization components (BatchNorm or LayerNorm), as aggregated features can vary widely in scale across reducers and hops. Compared with GNNs, FAFs typically favor larger learning rates, which can yield faster training, improved generalization via implicit regularization, and feature sparsity (Mohtashami et al., 2023; Sadrtdinov et al., 2024). Dropout levels, however, are broadly similar to those used for GNNs. This suggests that dropout's gains on these node-classification tasks are driven more by dataset properties than by the specifics of training graph convolutions, which nuances prior interpretations (Luo et al., 2025b).

## 5.2 ABLATIONS

**Ablation on single reducers.** Concatenating multiple aggregations has advantages and drawbacks. On the plus side, an MLP can learn to weight each reducer, removing the need to pick one per dataset. Because our individual reducers are not lossless, different datasets may favor different ones; moreover, adding informative, correlated covariates can improve robustness and reduce variance (Reddy et al., 2025). On the downside, concatenation increases input dimensionality, with

corresponding memory and optimization costs. Table 2 reports validation results when using a single aggregation at a time. Note that this resembles a simple feature selection over $FAF_4$. We keep the same hyperparameters as $FAF_4$ to isolate the effect of the aggregation choice, though the lower dimensionality could allow for alternative settings that further improve performance. Surprisingly, a single reducer often suffices, though the preferred choice varies by dataset. The mean is most frequently strongest, sometimes surpassing $FAF_4$ and $FAF_{mean,std}$—e.g., on Pubmed, where $FAF_{mean}$ outperforms the GNNs while other FAF variants do not. This may reflect optimization challenges from high-dimensional inputs or increased overfitting. Still, mean is not universally best: sum or max win on some datasets (e.g., Citeseer favors sum; Amazon-Ratings favors max). Combining reducers therefore remains beneficial when one wishes to avoid committing to a specific one a priori.

**Comparison with one-layer classifier and last hop.** Other simplifications of GNNs (Wu et al., 2019; Micheli & Tortorella, 2024) effectively fix the aggregation and train a single linear layer on the final-hop representation. In contrast, we concatenate representations from all hops and train a well-tuned MLP classifier. This choice is crucial for matching GNN performance. As shown in Table 7, MLPs consistently outperform a single linear layer applied to the same concatenated features, indicating that their nonlinearity and increased capacity are important to learn from multi-hop features. Moreover, Table 8 shows that only using the last hop lacks important information that is not transmitted across aggregations.

**Kolmogorov-Arnold aggregation.** Our hypothesis that Roman-Empire lags due to information loss is reinforced by a FAF variant that uses the Kolmogorov–Arnold (KA) function $\Phi$, which is theoretically lossless (Theorem 2). As we also exemplified in Fig. 3, in practice, KA is hard to use for classification. We observe in Table 9 that some datasets struggle to fit (lower training accuracy), while others show mild overfitting. Nevertheless, on Roman-Empire this variant attains a validation accuracy of $80.45 \pm 0.25$, the highest among all FAFs, suggesting that providing full neighborhood information helps close the gap on this task. This, in turn, highlights the need for benchmarks where predictive signal genuinely arises at distant hops in complex ways.

## 6 CONCLUSIONS

We have introduced Fixed Aggregation Features (FAFs), a non-learnable tabular mapping from local neighborhoods of graph features to univariate representations that an MLP can learn to classify. Our analysis shows that fixed, injective neighborhood aggregation functions exist, linking multiset expressivity to Kolmogorov–Arnold factorizations; thus learned message passing is not required for expressivity in theory. But in practice, common non-injective reducers (mean, sum, max, min) train more reliably, underscoring a gap between what is expressive in principle and what is reliably learnable. We also highlight the practical advantages of a tabular view, such as access to the rich tabular toolkit of interpretability and tuning, and isolated representation from inference so we can attribute gains or failures to the features themselves rather than to message-passing optimization.

On node classification datasets, FAFs are a strong baseline: they match or beat classic GNNs on many benchmarks, and trail only on two datasets needing longer-range interactions, where residualized GNNs help. Two ablations explain most gains: a well-tuned MLP beats a single linear classifier on top of FAF features, and concatenating all hop beats using only the last hop. This is consistent with later hops losing detail for these practical aggregation schemes. Surprisingly, two hops usually suffice, suggesting either limited signal in current benchmarks, or difficulty training deep GNNs to exploit more of it. While our theory carries over other downstream tasks, other benchmarks may surface different constraints that can alter the empirical outcomes.

Our findings have immediate implications. We recommend always including a tuned FAF baseline in future studies to calibrate what fixed aggregation alone can achieve; re-evaluating—and, when appropriate, retiring—datasets on which FAFs reach state-of-the-art performance; and developing benchmarks that genuinely require long-range dependencies and inter-hop dynamics. More broadly, we advocate for simplifying models and balancing expressiveness against optimizability, rather than assuming that extra parameters or higher expressiveness extract more relevant signal than simple baselines. Notably, several phenomena that are often blamed on graph architectures—overfitting, depth-related degradation, and sensitivity to dropout—also arise in tabular settings, indicating that some limitations may stem from dataset properties rather than the graph-aware architectures alone.

## REPRODUCIBILITY STATEMENT

Experimental details are provided in Appendix B, and further experimental results are included in Appendix C. Detailed train/validation/test performance across all epochs of the main runs can be found in Appendix D. The code for the experiments is attached as supplement.

## LLM STATEMENT

To improve fluency of the text sentence level, editing has been done using large language models.

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

# APPENDIX

## A STUDY OF REDUCERS FOR NEIGHBORHOOD AGGREGATION

### A.1 KOLMOGOROV-ARNOLD FUNCTION Φ AND ITS CONTINUOUS INVERSE

Here we showcase the behavior of $\Phi$ from the Kolmogorov-Arnold representation from Thm. 2. In Fig. 3a, we see the $\Phi$ image of two circles colored by their angle. Colors that were close together end up in separate parts of the Cantor set; for instance, oranges and reds, or purples and blues. In contrast, in Fig. 3b we see $\Phi^{-1}$ maps the Cantor set to the circles in such a way that all colors maintain their closeness.

If we use $\Phi$ as a fixed neighborhood aggregation, the classifier on top needs to learn to reverse it, therefore it is advantageous to have a continuous inverse. However, this does not give information about neighborhood distributions like the commonly used mean, sum, and max. In Fig. 3c we show the behavior of mean and std; mean gives approximate location but fuses together points that are very far apart. For instance, blues and reds have an unusually large first (but different) coordinate and are mapped to the center; however, this information can be recovered with std.

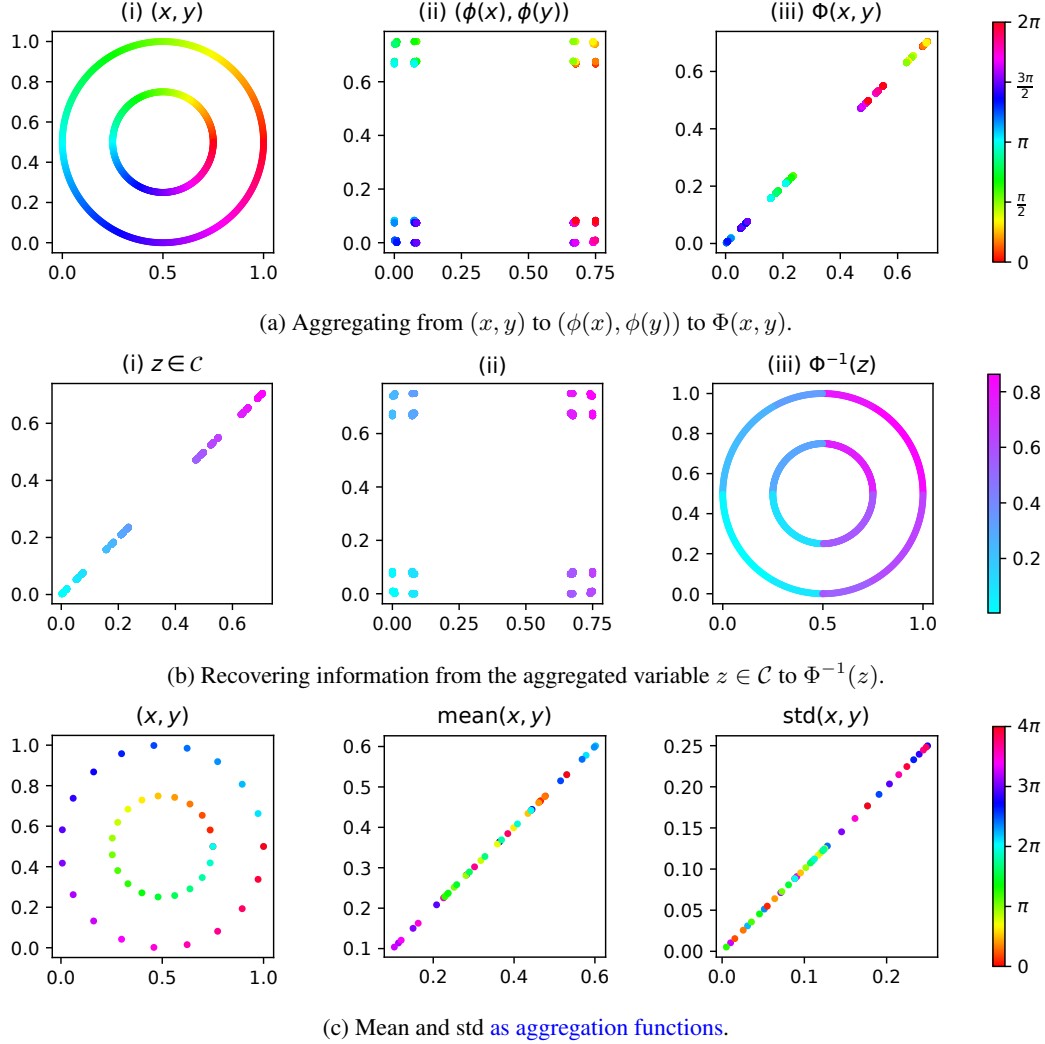

(a) Aggregating from $(x, y)$ to $(\phi(x), \phi(y))$ to $\Phi(x, y)$.

(b) Recovering information from the aggregated variable $z \in \mathcal{C}$ to $\Phi^{-1}(z)$.

(c) Mean and std as aggregation functions.

Figure 3: Functions $\Phi$ (Thm 2)—and its inverse—, mean and std. Circles and square-like panels (a.i, a.ii, b.iii, c.i) live in the 2D space, while segments and Cantor sets (a.iii, b.i, c.ii, c.iii) live in 1D. Colors in (a) and (c) are based on angles on 2D, while colors in (b) are based on position.

## A.2 PROOF OF MAIN THEOREM

For convenience, the following theorem restates Theorem 1 of the main paper.

**Theorem** (1-hop aggregation). *Assume the features $\mathcal{X}$ are orthogonal. Then, the function $h(X) = \sum_{x \in X} x$ defined on multisets $X \subseteq \mathcal{X}$ of bounded size is injective. Moreover, any multiset function $f$ can be decomposed as $f(X) = g\left(\sum_{x \in X} x\right)$ for some function g.*

*Proof.* Note that the multiset $X$ is fully characterized by the number $n_f$ of nodes in the set that have a feature $x_f$ for all possible features $x_f$. Our objective is to show that this information is contained in the aggregated form $h(X) = \sum_{x \in X} x$.

So let us assume that the features are orthogonal. Accordingly, the features $x_v$ of each node $v$ assume one of a finite number of possible states $x_1, \cdots x_n \in \mathbb{R}^{n_f}$ with $n_f \geq n$ and $x_i^T x_j = 0$ for any pair $i, j \in V$ with $i \neq j$. Note that the number of possible feature states $n$ must be finite even in an infinitely large graph, as long as the number of features are finite, i.e. $n_f < \infty$. Since the feature values must be pairwise orthogonal, there can maximally exist $n_f$ distinct feature vectors, as $n_f$ orthogonal vectors would form a basis of $\mathbb{R}^{n_f}$ and therefore an additional vector would become linearly dependent on the basis vectors.

So let us consider any of the possible feature states $x_f$. Then $x_f^T h(X) = \sum_{x \in X} x_f^T x = \sum_{x \in X, x = x_f} 1 = n_{x_f}$ counts the number of nodes in the set $S$ that have features $x_f$. Since this holds for all possible feature vectors $x_f$, all information about any multiset $X$ is preserved by $h(X)$.

Accordingly, we can write any multiset function $f(X) = (f(n_{x_1}), \cdots, f(n_{x_n}))$ (which transforms the feature counts) into a function $g$ that extracts first the count information from the sum $h(X)$. Concretely, we can define: $g(h(X))_f := f(n_{x_f}) = f(x_f^T h(X))$. $\square$

## A.3 LOSS OF INFORMATION OVER SECOND HOPS

We now explore an example of a computational tree of a node with two rounds of sum aggregation, and the qualitative kind of information that is lost from the first to the second hop. As shown by Xu et al. (2019) and generalized in Thm. 1, sum is injective over one-hot encoded features, but the second aggregation round sums features that are not necessarily orthogonal, and therefore loses neighborhood information. The computational tree and calculation of hops are displayed in Figure 4.

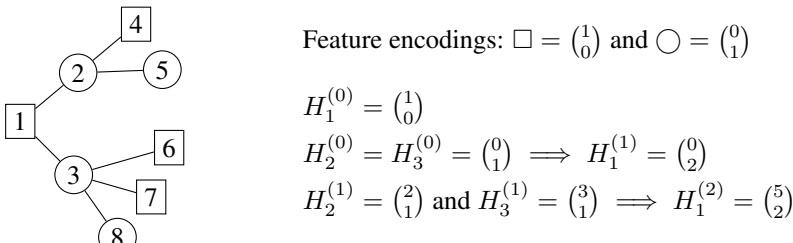

Feature encodings: $\square = \binom{1}{0}$ and $\bigcirc = \binom{0}{1}$

$H_1^{(0)} = \binom{1}{0}$

$H_2^{(0)} = H_3^{(0)} = \binom{0}{1} \implies H_1^{(1)} = \binom{0}{2}$

$H_2^{(1)} = \binom{2}{1}$ and $H_3^{(1)} = \binom{3}{1} \implies H_1^{(2)} = \binom{5}{2}$

Figure 4: Example of a two-hop neighborhood with one-hot encoded features and sum aggregation.

Given the previous hops $H_1^{(0)} = \binom{1}{0}$ and $H_1^{(1)} = \binom{0}{2}$, and the second hop $H_1^{(2)} = \binom{5}{2}$, what other combinations of two-hop neighborhoods can there be for node 1? Apart from itself, node 1's two-hop neighbors are in a $\binom{3}{2}$ feature ratio. However, we have lost the ability to recognize a) how many belong to each of its one-hop neighbor; and b) the distribution or homogeneity of each neighborhood. In reality, these 5 nodes are approximately spread out in number and distribution across the one-hop neighbors 2 and 3. But alternatively, all 5 nodes could have belonged to node 2, or all squared nodes could have belonged to node 3.

Note that, without previous hops $H_1^{(0)}$ and $H_1^{(1)}$, we cannot even distinguish node 1's original features, nor distinguish its presence as a neighbor in its one-hop neighbors. Therefore, concatenating all hops is advantageous.

For completeness, we also include the calculations for mean aggregation, which are qualitatively similar to the sum in this case.

$$H_1^{(0)} = \binom{1}{0} \; ; \quad H_2^{(0)} = H_3^{(0)} = \binom{0}{1} \quad \implies H_1^{(1,m)} = \binom{0}{1}$$

$$H_2^{(1,m)} = \binom{2/3}{1/3} \text{ and } H_3^{(1.m)} = \binom{3/4}{1/4} \quad \implies H_1^{(2,m)} = \binom{17/24}{7/24}$$

A good lossless aggregation scheme should take all possible second-hop neighborhood distributions and map them to values that would not lose information when aggregated. For instance, choosing $a, b$ such that $\binom{2}{1} \to a$ from node 2, and $\binom{3}{1} \to b$ from node 3, so that, $H_1^{(2)} = a + b$ could recover both values separately. Naturally, mapping them to one-hot encodings per distribution would suffice, but it would grow exponentially. This opens the door for better suitable fixed aggregations, or perhaps other kinds of learnable aggregation beyond current understanding of message passing.

## B  EXPERIMENTAL DETAILS

### B.1  DATASET DETAILS

Datasets are taken directly from the setup of Luo et al. (2024), which includes varied node classification datasets. Here in Table 3 we include for completeness the same overview of these benchmarks.

Table 3: Details of the node classification datasets.

| Dataset | Type | # Nodes | # Edges | # Features | Classes | Metric | Origin |
|---|---|---|---|---|---|---|---|
| Cora | Homophily | 2,708 | 5,278 | 1,433 | 7 | Accuracy | (McCallum et al., 2000) |
| CiteSeer | Homophily | 3,327 | 4,522 | 3,703 | 6 | Accuracy | (Sen et al., 2008) |
| PubMed | Homophily | 19,717 | 44,324 | 500 | 3 | Accuracy | (Namata et al., 2012) |
| Computer | Homophily | 13,752 | 245,861 | 767 | 10 | Accuracy | (Shchur et al., 2018) |
| Photo | Homophily | 7,650 | 119,081 | 745 | 8 | Accuracy | (Shchur et al., 2018) |
| CS | Homophily | 18,333 | 81,894 | 6,805 | 15 | Accuracy | (Shchur et al., 2018) |
| Physics | Homophily | 34,493 | 247,962 | 8,415 | 5 | Accuracy | (Shchur et al., 2018) |
| WikiCS | Homophily | 11,701 | 216,123 | 300 | 10 | Accuracy | (Mernyei & Cangea, 2020) |
| Squirrel | Heterophily | 2,223 | 46,998 | 2,089 | 5 | Accuracy | (Rozemberczki et al., 2021) |
| Chameleon | Heterophily | 890 | 8,854 | 2,325 | 5 | Accuracy | (Rozemberczki et al., 2021) |
| Roman-Empire | Heterophily | 22,662 | 32,927 | 300 | 18 | Accuracy | (Platonov et al., 2023b) |
| Amazon-Ratings | Heterophily | 24,492 | 93,050 | 300 | 5 | Accuracy | (Platonov et al., 2023b) |
| Minesweeper | Heterophily | 10,000 | 39,402 | 7 | 2 | ROC-AUC | (Platonov et al., 2023b) |
| Questions | Heterophily | 48,921 | 153,540 | 301 | 2 | ROC-AUC | (Platonov et al., 2023b) |

### B.2  HYPERPARAMETERS

Each experiment is run on an NVIDIA A100 GPU. The setup is taken from Luo et al. (2024). That is, for a maximum of 2500 epochs, we tune the following parameters:

1. DROPOUT ∈ (0.0 0.2 0.3 0.5 0.7)
2. LR ∈ (0.01 0.005 0.001 0.0001)
3. NORMALIZATION ∈ (ln bn none)
4. HIDDEN CHANNELS ∈ (64 256 512)

While Luo et al. (2024) includes weight decay as a hyperparameter, there are no concrete ranges specified for it. Therefore, we tune all the different values from the best runs of the given datasets:

5. WEIGHT DECAY ∈ (0.0 1e-2 1e-3 5e-4 5e-5).

Moreover, Luo et al. (2024) tunes the local layers from 1 to 10 or 15. We instead take for each dataset the same value that they have found best for the GNNs, and use it to construct our fixed aggregation features up to that depth. In some cases where there are too many features, we restrict the depth to a smaller value, thus including a strict subset of features instead. This serves as an ad hoc feature selection to reduce overfitting.

We include as hyperparameter the MLP depth. We also include results for MLP = 1 in Table 7.

Table 4: Best hyperparameters for FAF. Classic GNNs are taken from (Luo et al., 2024).

| Dataset | Model | dropout | lr | bn | ln | hidden channels | wd | hops | mlp layers | res |
|---|---|---|---|---|---|---|---|---|---|---|
| computer | GCN | 0.5 | 0.001 | 0 | 1 | 512 | 5e-05 | 3 | 0 | 0 |
| | GAT | 0.5 | 0.001 | 0 | 1 | 64 | 5e-05 | 2 | 0 | 0 |
| | SAGE | 0.3 | 0.001 | 0 | 1 | 64 | 5e-05 | 4 | 0 | 0 |
| | FAF | 0.7 | 0.005 | 1 | 0 | 256 | 5e-05 | 2 | 2 | 0 |
| photo | GCN | 0.5 | 0.001 | 0 | 1 | 256 | 5e-05 | 6 | 0 | 1 |
| | GAT | 0.5 | 0.001 | 0 | 1 | 64 | 5e-05 | 3 | 0 | 1 |
| | SAGE | 0.2 | 0.001 | 0 | 1 | 64 | 5e-05 | 6 | 0 | 1 |
| | FAF | 0.5 | 0.005 | 1 | 0 | 256 | 0.0005 | 4 | 2 | 0 |
| ratings | GCN | 0.5 | 0.001 | 1 | 0 | 512 | 0 | 4 | 0 | 1 |
| | GAT | 0.5 | 0.001 | 1 | 0 | 512 | 0 | 4 | 0 | 1 |
| | SAGE | 0.5 | 0.001 | 1 | 0 | 512 | 0 | 9 | 0 | 1 |
| | FAF | 0.2 | 0.001 | 1 | 0 | 256 | 0 | 3 | 2 | 0 |
| chameleon | GCN | 0.2 | 0.005 | 0 | 0 | 512 | 0.001 | 5 | 0 | 0 |
| | GAT | 0.7 | 0.01 | 1 | 0 | 256 | 0.001 | 2 | 0 | 1 |
| | SAGE | 0.7 | 0.01 | 1 | 0 | 256 | 0.001 | 4 | 0 | 1 |
| | FAF | 0.3 | 0.001 | 1 | 0 | 512 | 0.01 | 5 | 5 | 0 |
| citeseer | GCN | 0.5 | 0.001 | 0 | 0 | 512 | 0.01 | 2 | 0 | 0 |
| | GAT | 0.5 | 0.001 | 0 | 0 | 256 | 0.01 | 3 | 0 | 1 |
| | SAGE | 0.2 | 0.001 | 0 | 0 | 512 | 0.01 | 3 | 0 | 0 |
| | FAF | 0 | 0.005 | 0 | 1 | 512 | 0.001 | 2 | 3 | 0 |
| coauthor-cs | GCN | 0.3 | 0.001 | 0 | 1 | 512 | 0.0005 | 2 | 0 | 1 |
| | GAT | 0.3 | 0.001 | 0 | 1 | 256 | 0.0005 | 1 | 0 | 1 |
| | SAGE | 0.5 | 0.001 | 0 | 1 | 512 | 0.0005 | 2 | 0 | 1 |
| | FAF | 0.2 | 0.005 | 1 | 0 | 64 | 0.01 | 2 | 2 | 0 |
| coauthor-physics | GCN | 0.3 | 0.001 | 0 | 1 | 64 | 0.0005 | 2 | 0 | 1 |
| | GAT | 0.7 | 0.001 | 1 | 0 | 256 | 0.0005 | 2 | 0 | 1 |
| | SAGE | 0.7 | 0.001 | 1 | 0 | 64 | 0.0005 | 2 | 0 | 1 |
| | FAF | 0 | 0.001 | 1 | 0 | 512 | 0.001 | 1 | 2 | 0 |
| cora | GCN | 0.7 | 0.001 | 0 | 0 | 512 | 0.0005 | 3 | 0 | 0 |
| | GAT | 0.2 | 0.001 | 0 | 0 | 512 | 0.0005 | 3 | 0 | 1 |
| | SAGE | 0.7 | 0.001 | 0 | 0 | 256 | 0.0005 | 3 | 0 | 0 |
| | FAF | 0.7 | 0.01 | 0 | 1 | 512 | 0.01 | 3 | 3 | 0 |
| minesweeper | GCN | 0.2 | 0.01 | 1 | 0 | 64 | 0 | 12 | 0 | 1 |
| | GAT | 0.2 | 0.01 | 1 | 0 | 64 | 0 | 15 | 0 | 1 |
| | SAGE | 0.2 | 0.01 | 1 | 0 | 64 | 0 | 15 | 0 | 1 |
| | FAF | 0.2 | 0.01 | 1 | 0 | 64 | 0 | 4 | 12 | 0 |
| pubmed | GCN | 0.7 | 0.005 | 0 | 0 | 256 | 0.0005 | 2 | 0 | 0 |
| | GAT | 0.5 | 0.01 | 0 | 0 | 512 | 0.0005 | 2 | 0 | 0 |
| | SAGE | 0.7 | 0.005 | 0 | 0 | 512 | 0.0005 | 4 | 0 | 0 |
| | FAF | 0.7 | 0.01 | 0 | 1 | 64 | 0 | 4 | 2 | 0 |
| questions | GCN | 0.3 | 3e-05 | 0 | 0 | 512 | 0 | 10 | 0 | 1 |
| | GAT | 0.2 | 3e-05 | 0 | 1 | 512 | 0 | 3 | 0 | 1 |
| | SAGE | 0.2 | 3e-05 | 0 | 1 | 512 | 0 | 6 | 0 | 0 |
| | FAF | 0.2 | 0.005 | 1 | 0 | 512 | 0.01 | 4 | 3 | 0 |
| roman-empire | GCN | 0.5 | 0.001 | 1 | 0 | 512 | 0 | 9 | 0 | 1 |
| | GAT | 0.3 | 0.001 | 1 | 0 | 512 | 0 | 10 | 0 | 1 |
| | SAGE | 0.3 | 0.001 | 1 | 0 | 256 | 0 | 9 | 0 | 0 |
| | FAF | 0.7 | 0.01 | 1 | 0 | 256 | 0 | 2 | 3 | 0 |
| squirrel | GCN | 0.7 | 0.01 | 1 | 0 | 256 | 0.0005 | 4 | 0 | 1 |
| | GAT | 0.5 | 0.005 | 1 | 0 | 512 | 0.0005 | 7 | 0 | 1 |
| | SAGE | 0.7 | 0.01 | 1 | 0 | 256 | 0.0005 | 3 | 0 | 1 |
| | FAF | 0.7 | 0.01 | 1 | 0 | 512 | 0.01 | 4 | 5 | 0 |
| wikics | GCN | 0.5 | 0.001 | 0 | 1 | 256 | 0 | 3 | 0 | 0 |
| | GAT | 0.7 | 0.001 | 0 | 1 | 512 | 0 | 2 | 0 | 1 |
| | SAGE | 0.7 | 0.001 | 0 | 1 | 256 | 0 | 2 | 0 | 0 |
| | FAF | 0.7 | 0.01 | 1 | 0 | 64 | 0.001 | 2 | 2 | 0 |

6. MLP LAYERS $\in$ (2 3 5).

On the other hand, we do not include linear residual connections, as these are used to bypass the convolutional layers in the classical GNNs. This creates a direct difference on the two datasets that most benefit from this component, Minesweeper and Roman-Empire.

Table 5: Empirical training time in seconds of FAF and GNN models, averaged over runs.

| Dataset | GCN | GAT | SAGE | $FAF_4$ | $FAF_2$ | $FAF_1$ | MLP |
|---|---|---|---|---|---|---|---|
| computer | 127.30 ± 0.58 | 29.33 ± 3.22 | 45.33 ± 0.58 | 42.67 ± 0.58 | 25.00 ± 0.00 | 17.00 ± 0.00 | 9.33 ± 0.58 |
| photo | 82.00 ± 0.00 | 26.00 ± 0.00 | 33.33 ± 0.58 | 66.67 ± 0.58 | 36.00 ± 0.00 | 21.33 ± 0.58 | 7.33 ± 0.58 |
| ratings | 140.30 ± 0.58 | 161.30 ± 0.58 | 330.00 ± 0.00 | 46.00 ± 0.00 | 26.00 ± 0.00 | 17.33 ± 0.58 | 10.00 ± 0.00 |
| chameleon | 22.30 ± 0.48 | 15.20 ± 0.42 | 18.00 ± 0.00 | 44.50 ± 0.97 | 22.50 ± 0.53 | 15.10 ± 0.32 | 10.00 ± 0.00 |
| citeseer | 16.20 ± 0.45 | 20.00 ± 0.00 | 26.00 ± 0.00 | 62.80 ± 1.79 | 36.20 ± 0.45 | 25.80 ± 1.79 | 13.60 ± 0.55 |
| coauthor-cs | 157.00 ± 0.00 | 70.33 ± 0.58 | 301.30 ± 20.50 | 296.30 ± 1.16 | 159.00 ± 0.00 | 91.00 ± 0.00 | 26.33 ± 0.58 |
| coauthor-physics | 65.33 ± 0.58 | 190.00 ± 0.00 | 400.30 ± 0.58 | 2383.00 ± 0.58 | 1004.00 ± 1.00 | 532.70 ± 0.58 | 183.00 ± 0.00 |
| cora | 16.40 ± 0.89 | 20.20 ± 0.45 | 11.20 ± 0.45 | 46.80 ± 0.45 | 26.00 ± 0.00 | 17.00 ± 0.00 | 8.20 ± 0.45 |
| minesweeper | 69.67 ± 0.58 | 100.30 ± 0.58 | 68.67 ± 2.08 | 19.00 ± 0.00 | 20.00 ± 3.46 | 19.00 ± 0.00 | 18.00 ± 1.73 |
| pubmed | 20.20 ± 0.45 | 42.00 ± 0.00 | 78.20 ± 0.45 | 34.00 ± 0.00 | 19.20 ± 0.45 | 12.00 ± 0.00 | 6.20 ± 0.45 |
| questions | 650.00 ± 0.00 | 258.00 ± 1.00 | 363.70 ± 0.58 | 180.70 ± 0.58 | 122.70 ± 0.58 | 94.00 ± 0.00 | 64.67 ± 1.16 |
| roman-empire | 240.30 ± 0.58 | 294.30 ± 0.58 | 93.00 ± 0.00 | 38.67 ± 2.89 | 23.33 ± 0.58 | 18.00 ± 0.00 | 14.67 ± 2.89 |
| squirrel | 29.00 ± 0.00 | 87.10 ± 0.32 | 24.70 ± 1.89 | 43.30 ± 0.48 | 25.20 ± 0.42 | 18.00 ± 0.00 | 11.70 ± 0.48 |
| wikics | 60.00 ± 3.46 | 97.33 ± 0.58 | 27.33 ± 0.58 | 10.00 ± 0.00 | 7.33 ± 0.58 | 7.00 ± 0.00 | 7.00 ± 0.00 |

In Table 4 we include the best hyperparameter choices for the four models: GCN, GATv2, Graph-SAGE, and $FAF_4$, the results of which are in Table 2. We run baselines directly from the setup of Luo et al. (2024), and we sweep $FAF_4$ with the same ranges included in their work. Each dataset has a specific number of splits given by their setup (from 3 to 10), which we then average.

We also include in Table 5 the training runtime of our algorithms, including $FAF_4$ and its variants, grouped by the number of reducers—as MLPs with the same input width will have the same training time. Note that all runs have the same number of epochs (2500), as in the original setup, and all datasets match the number of runs of the setup. In general, MLPs are more efficient than MPGNNs, as backpropagation over message-passing is costly. However, we increase the number of features in the data—depending on the aggregation depth and number of reducers—so for some datasets with many features the improvement is not necessarily observed. A way to reduce this overhead may be to apply dimensionality reduction to the tabular FAF representation.

### B.3 FAFs BEYOND NODE CLASSIFICATION

Our theory applies to any task that learns multiset functions over neighborhoods. In our experiments, we focus on node classification for two main reasons. First, these are the benchmarks on which GNNs have been shown to be competitive with more complex architectures in Luo et al. (2024), so they are amenable to simple models for which we have strong, well-tuned hyperparameters. Second, node-classification datasets typically provide rich features that depend on neighborhood distributions. Thus, non-injective but commonly used reducers such as mean and sum still convey highly informative distributional signals. Regarding inductive settings, they would require computing the new aggregation rounds at test time. We would not have access to the test node features when precomputing training aggregations. Otherwise, our approach is just as feasible as in the transductive case.

## C ADDITIONAL EXPERIMENTS

In Table 6 we include results for different number of hops concatenated as features for $FAF_4$. Note that one hop already gives much of the information, and two hops often give the best performance.

In Table 7 we show the performance of two classifiers on the same $FAF_4$ features: one linear layer and a multilayer perceptron—this being our choice for other experiments.

In Table 8 we ablate on using only the last hop as features to an MLP, or using the last hop concatenated to the original features. This mimics the choice of directly freezing a GNN and using its output as features to an (often linear) classifier.

Table 9 shows the last epoch training accuracy and best epoch validation accuracy of using the Kolmogorov-Arnold function $\Phi$ as a reducer for FAF. Following Corso et al. (2020), we make it act on multisets by sorting, which we fix by the given data order.

In Table 10 we include results on rewiring the input graph by deleting edges based on pairwise cosine similarity. REW includes hop-wise features where negative similarity neighbors are set to

0. SP includes hop-wise features where positive and negative similarity neighbors are aggregated in different features and concatenated together.

Table 6: Increasing number of concatenated hops in $FAF_4$, compared to the best GNN (Classic).

| Dataset | computer | photo | ratings | chameleon | citeseer | coauthor-cs | coauthor-physics |
|---------|----------|-------|---------|-----------|----------|-------------|------------------|
| Classic | 92.86 ± 0.06 | 95.93 ± 0.15 | 55.90 ± 0.54 | 48.15 ± 2.35 | 70.36 ± 0.09 | 95.51 ± 0.04 | 97.16 ± 0.07 |
| FAF+0 | 87.89 ± 0.13 | 93.33 ± 0.07 | 48.98 ± 0.72 | 41.43 ± 1.77 | 53.80 ± 0.82 | 93.70 ± 0.07 | 95.89 ± 0.02 |
| FAF+1 | 92.53 ± 0.08 | 96.14 ± 0.07 | 54.17 ± 0.14 | 46.91 ± 1.43 | 65.52 ± 0.64 | 94.84 ± 0.08 | 96.83 ± 0.01 |
| FAF+2 | 93.05 ± 0.04 | 96.23 ± 0.08 | 55.02 ± 0.67 | 47.30 ± 1.76 | 67.28 ± 0.64 | 94.93 ± 0.07 | 96.63 ± 0.02 |
| FAF+4 | 93.04 ± 0.10 | 96.34 ± 0.07 | 55.08 ± 0.19 | 48.42 ± 2.22 | 50.52 ± 3.84 | 94.88 ± 0.07 | 96.63 ± 0.04 |
| FAF+8 | 92.96 ± 0.13 | 96.21 ± 0.07 | 55.26 ± 0.30 | 48.74 ± 1.69 | 40.72 ± 2.26 | 94.92 ± 0.09 | - |

| Dataset | cora | minesweeper | pubmed | questions | roman-empire | squirrel | wikics |
|---------|------|-------------|--------|-----------|--------------|----------|--------|
| Classic | 81.32 ± 0.41 | 97.68 ± 0.63 | 79.08 ± 0.23 | 78.63 ± 0.23 | 91.14 ± 0.58 | 44.88 ± 1.27 | 82.38 ± 0.57 |
| FAF+0 | 62.68 ± 1.15 | 51.12 ± 0.93 | 71.12 ± 0.52 | 71.58 ± 1.46 | 66.28 ± 0.27 | 40.57 ± 0.92 | 74.86 ± 0.33 |
| FAF+1 | 82.16 ± 0.33 | 87.65 ± 0.47 | 78.24 ± 0.36 | 77.44 ± 1.07 | 77.36 ± 0.55 | 47.26 ± 1.31 | 81.37 ± 0.51 |
| FAF+2 | 82.84 ± 0.38 | 89.48 ± 1.08 | 78.52 ± 0.41 | 79.71 ± 0.86 | 78.68 ± 0.19 | 47.18 ± 1.44 | 81.92 ± 0.43 |
| FAF+4 | 81.80 ± 0.35 | 89.63 ± 1.02 | 79.08 ± 0.36 | 79.67 ± 0.89 | 77.48 ± 0.17 | 47.61 ± 1.43 | 81.73 ± 0.53 |
| FAF+8 | 74.28 ± 0.23 | 89.10 ± 1.03 | 76.80 ± 0.20 | 79.94 ± 0.88 | 75.08 ± 0.23 | 47.95 ± 1.36 | 81.58 ± 0.63 |

Table 7: Comparison of 1 linear layer (1L) versus multiple layers (MLP) as the classifier over $FAF_4$.

| Dataset | computer | photo | ratings | chameleon | citeseer | coauthor-cs | coauthor-physics |
|---------|----------|-------|---------|-----------|----------|-------------|------------------|
| FAF+MLP | 93.05 ± 0.04 | 96.34 ± 0.07 | 55.53 ± 0.43 | 48.51 ± 2.31 | 67.28 ± 0.64 | 94.93 ± 0.07 | 96.83 ± 0.01 |
| FAF+1L | 91.50 ± 0.08 | 96.01 ± 0.00 | 47.65 ± 0.75 | 47.11 ± 2.68 | 66.76 ± 0.99 | 93.32 ± 0.05 | 96.62 ± 0.01 |

| Dataset | cora | minesweeper | pubmed | questions | roman-empire | squirrel | wikics |
|---------|------|-------------|--------|-----------|--------------|----------|--------|
| FAF+MLP | 82.84 ± 0.43 | 89.63 ± 1.03 | 79.08 ± 0.36 | 79.53 ± 1.12 | 78.68 ± 0.19 | 47.31 ± 1.39 | 81.92 ± 0.43 |
| FAF+1L | 81.08 ± 0.41 | 88.99 ± 0.76 | 78.00 ± 0.51 | 77.22 ± 1.04 | 75.56 ± 0.28 | 45.89 ± 1.75 | 81.25 ± 0.96 |

Table 8: Using only the last hop ($H^{(K)}$), that and original features ($H^{(0)\oplus(K)}$), and all hops ($FAF_4$).

| Dataset | computer | photo | ratings | chameleon | citeseer | coauthor-cs | coauthor-physics |
|---------|----------|-------|---------|-----------|----------|-------------|------------------|
| $FAF_4$ | 93.05 ± 0.04 | 96.34 ± 0.07 | 55.53 ± 0.43 | 48.51 ± 2.31 | 67.28 ± 0.64 | 94.93 ± 0.07 | 96.83 ± 0.01 |
| $H^{(0)\oplus(K)}$ | 92.35 ± 0.08 | 95.53 ± 0.04 | 54.95 ± 0.23 | 48.16 ± 2.61 | 66.80 ± 0.42 | 95.13 ± 0.02 | 96.65 ± 0.01 |
| $H^{(K)}$ | 91.67 ± 0.04 | 92.18 ± 0.04 | 49.64 ± 0.40 | 49.18 ± 2.14 | 66.68 ± 0.46 | 92.83 ± 0.10 | 96.22 ± 0.06 |

| Dataset | cora | minesweeper | pubmed | questions | roman-empire | squirrel | wikics |
|---------|------|-------------|--------|-----------|--------------|----------|--------|
| $FAF_4$ | 82.84 ± 0.43 | 89.63 ± 1.03 | 79.08 ± 0.36 | 79.53 ± 1.12 | 78.68 ± 0.19 | 47.31 ± 1.39 | 81.92 ± 0.43 |
| $H^L$ | 82.12 ± 0.52 | 69.40 ± 1.50 | 79.24 ± 0.36 | 78.14 ± 0.93 | 55.43 ± 0.51 | 46.10 ± 1.46 | 80.14 ± 0.67 |
| $H^{(0)\oplus(K)}$ | 81.80 ± 0.28 | 73.30 ± 1.26 | 79.64 ± 0.38 | 79.29 ± 0.27 | 76.42 ± 0.13 | 46.41 ± 1.84 | 81.22 ± 0.52 |
| $H^{(K)}$ | 82.12 ± 0.52 | 69.40 ± 1.50 | 79.24 ± 0.36 | 78.14 ± 0.93 | 55.43 ± 0.51 | 46.10 ± 1.46 | 80.14 ± 0.67 |

Table 9: (Last epoch) training and (best) validation accuracy of the KA function $\Phi$ from Thm. 2.

| Dataset | computer | photo | ratings | chameleon | citeseer | coauthor-cs | coauthor-physics |
|---------|----------|-------|---------|-----------|----------|-------------|------------------|
| $FAF_{KA}$ (train) | 94.73 ± 0.12 | 99.65 ± 0.09 | 99.92 ± 0.02 | 96.35 ± 3.44 | 70.17 ± 41.23 | 99.49 ± 0.22 | 99.84 ± 0.28 |
| $FAF_{KA}$ (val) | 87.88 ± 0.19 | 93.40 ± 0.07 | 51.97 ± 0.12 | 41.65 ± 1.91 | 55.76 ± 1.07 | 93.66 ± 0.08 | 95.90 ± 0.05 |

| Dataset | cora | minesweeper | pubmed | questions | roman-empire | squirrel | wikics |
|---------|------|-------------|--------|-----------|--------------|----------|--------|
| $FAF_{KA}$ (train) | 14.29 ± 0.00 | 51.68 ± 0.44 | 33.33 ± 0.00 | 99.85 ± 0.09 | 86.41 ± 0.23 | 39.30 ± 2.30 | 97.64 ± 1.09 |
| $FAF_{KA}$ (val) | 29.56 ± 0.79 | 51.61 ± 0.42 | 42.80 ± 0.76 | 74.28 ± 1.95 | 80.45 ± 0.25 | 40.43 ± 1.10 | 77.38 ± 0.88 |

Table 10: Feature augmentations based on similarity-based rewiring (REW) and computational graph splitting (SP) on a subset of datasets that benefit from mean aggregation.

| Dataset | computer | photo | chameleon | citeseer | cora | pubmed | wikics |
|---|---|---|---|---|---|---|---|
| $\text{FAF}_{mean}$ | 93.16 ± 0.04 | **96.06 ± 0.10** | 47.99 ± 2.02 | 66.92 ± 0.87 | 83.28 ± 0.30 | 81.16 ± 0.97 | 81.58 ± 0.46 |
| $\text{REW}_{mean}$ | 93.25 ± 0.08 | 95.90 ± 0.04 | 43.94 ± 2.39 | 66.92 ± 0.78 | 82.36 ± 0.17 | 80.80 ± 0.42 | 82.44 ± 0.56 |
| $\text{SP}_{mean}$ | 93.20 ± 0.11 | 95.97 ± 0.10 | 43.71 ± 1.71 | 67.32 ± 0.99 | 81.80 ± 0.24 | 80.84 ± 0.62 | 82.38 ± 0.46 |
| $\text{FAF}_{mean}+\text{REW}_{mean}$ | **93.33 ± 0.12** | 96.03 ± 0.04 | 48.26 ± 1.66 | 67.48 ± 0.36 | 83.20 ± 0.20 | 80.84 ± 0.52 | **82.46 ± 0.48** |
| $\text{FAF}_{mean}+\text{SP}_{mean}$ | 93.31 ± 0.10 | **96.06 ± 0.04** | 47.99 ± 1.83 | 67.88 ± 0.33 | 83.52 ± 0.36 | 81.24 ± 0.43 | 82.43 ± 0.51 |

In Figure 5 we show two more plots of feature importance using SHAP (Lundberg & Lee, 2017) for Pubmed and Amazon-Ratings, on the MLP over single-reducer FAFs. Features are sorted by global importance and broken down over the different hops by color. While the implementation of SHAP on MLPs used (GradientExplainer) relies on local linearization and often assumes input feature independence, the explanations still reveal informative qualitative patterns. In Pubmed, feature 346 is most important at hops 1 and 2, and remains second at hops 0 and 4, whereas the most important base feature (205) contributes little at other hops. By contrast, in Amazon-Ratings, importance is more evenly distributed across features and hops.

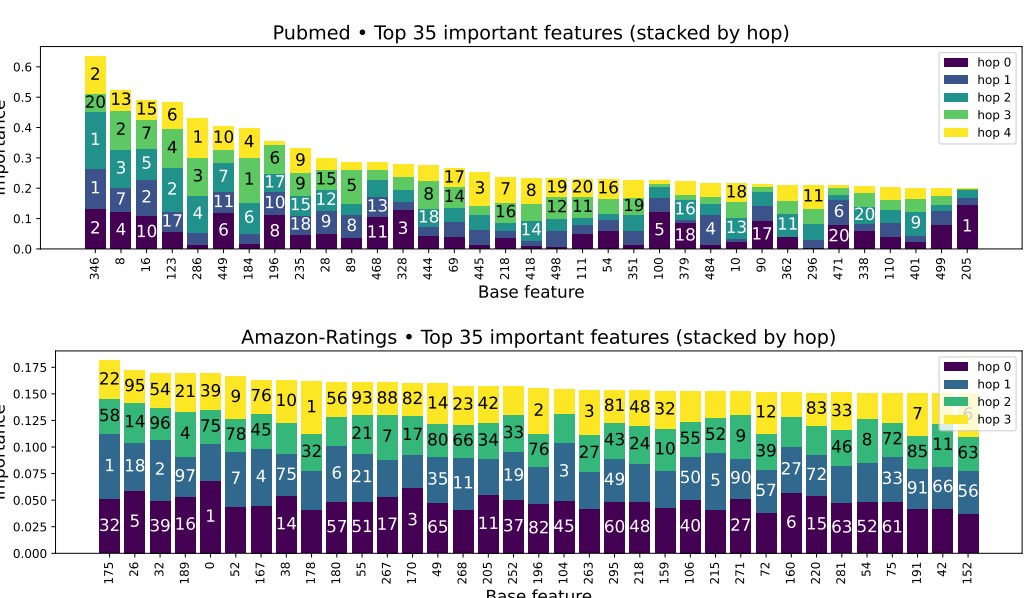

Figure 5: SHAP feature importance for Pubmed and Amazon-Ratings. The base features are ranked according to the sum of their importance values across hops. Numbers on the stacked bars indicate the ranking of that particular feature on that particular hop.

## D    TRAINING, VALIDATION, AND TEST ACCURACY CURVES

In this section we compare training MLPs on FAF features to training GCNs, by tracking train/validation/test accuracy over epochs (Figure 6). Below we summarize the behaviors on datasets where differences arise between the two methods:

- Amazon-Computer (6a) and Amazon-Photo (6b) behave similarly, but GCNs are more unstable.

- FAF for Chameleon (6d) has much better training accuracy but similar generalization; in contrast, GCN for Squirrel (6m) has much better training, but slightly worse generalization than FAF.

- Citeseer (6e) with FAF breaks at the end of training, which indicates instability. However, this could be overcome with standard learning rate schedules.

- Coauthor-CS (6f) and Coauthor-Physics (6g) have dips in all metrics for both models.

- Questions (6f) with FAF is more (locally) unstable but also more stationary and does not degrade performance later on.

- As mentioned in the main text, Minesweeper (6i) and Roman-Empire (6l) are the two datasets that seem to truly lose neighborhood information with FAF.

### D.1    TEST ACCURACY OF MAIN RESULTS

Here in Table 11 we report the test accuracy of the main FAF variants of our experimental results (§ 5) where in Table 1 we only have the best validation FAF's test results, and in Table 2 we show validation results. FAFs in all datasets are $\pm 1\%$ away from the best classic GNN, except for those already mentioned in the main text (Citeseer, Cora, Roman-Empire, and Minesweeper).

Table 11: Test accuracy on 14 node classification benchmarks: FAFs+MLP against classic GNNs. Validation accuracy is shown in Table 2.

| Dataset | computer | photo | ratings | chameleon | citeseer | coauthor-cs | coauthor-physics |
|---|---|---|---|---|---|---|---|
| GCN | 93.58 ± 0.44 | 95.77 ± 0.27 | 53.86 ± 0.48 | 44.62 ± 4.50 | **72.72 ± 0.45** | 95.73 ± 0.15 | **97.47 ± 0.08** |
| GAT | 93.91 ± 0.22 | 96.45 ± 0.37 | **55.51 ± 0.55** | 42.90 ± 5.47 | 71.82 ± 0.65 | 96.14 ± 0.08 | 97.12 ± 0.13 |
| SAGE | 93.31 ± 0.17 | 96.17 ± 0.44 | 55.26 ± 0.27 | 43.11 ± 4.73 | 71.82 ± 0.81 | **96.21 ± 0.10** | 97.10 ± 0.09 |
| MLP | 87.75 ± 0.42 | 93.62 ± 0.36 | 49.04 ± 0.39 | 38.59 ± 3.29 | 57.22 ± 2.25 | 93.80 ± 0.19 | 96.02 ± 0.16 |
| FAF$_{\text{bestval}}$ | **94.01 ± 0.21** | 96.54 ± 0.13 | 55.09 ± 0.24 | 42.96 ± 2.45 | 70.48 ± 1.24 | 95.37 ± 0.17 | 97.05 ± 0.18 |
| FAF$_4$ | 93.75 ± 0.04 | 96.54 ± 0.13 | 54.42 ± 0.45 | 42.96 ± 2.45 | 69.42 ± 1.32 | 95.33 ± 0.20 | 96.96 ± 0.09 |
| FAF$_{\text{mean,std}}$ | 94.00 ± 0.25 | 96.30 ± 0.23 | 54.73 ± 0.22 | **45.13 ± 3.42** | 67.90 ± 0.95 | 95.34 ± 0.14 | 96.93 ± 0.04 |
| FAF$_{\text{mean}}$ | **94.01 ± 0.21** | **96.71 ± 0.16** | 53.12 ± 0.44 | 43.21 ± 2.24 | 66.82 ± 1.74 | 95.37 ± 0.17 | 97.05 ± 0.18 |
| FAF$_{\text{max,std}}$ | 93.60 ± 0.25 | 96.01 ± 0.41 | 55.09 ± 0.24 | 43.20 ± 2.42 | 67.18 ± 0.88 | 95.53 ± 0.10 | 96.61 ± 0.04 |
| FAF$_{\text{max}}$ | 92.98 ± 0.22 | 96.12 ± 0.10 | 54.79 ± 0.15 | 42.15 ± 3.19 | 67.52 ± 0.40 | 95.55 ± 0.08 | 96.84 ± 0.13 |
| FAF$_{\text{sum}}$ | 91.77 ± 0.24 | 95.08 ± 0.61 | 53.44 ± 0.18 | 39.63 ± 2.90 | 70.48 ± 1.24 | 95.08 ± 0.12 | 96.86 ± 0.06 |
| FAF$_{\text{std}}$ | 93.54 ± 0.26 | 96.17 ± 0.10 | 54.77 ± 0.14 | 42.68 ± 2.75 | 62.70 ± 1.18 | 95.77 ± 0.12 | 96.97 ± 0.09 |

| Dataset | cora | minesweeper | pubmed | questions | roman-empire | squirrel | wikics |
|---|---|---|---|---|---|---|---|
| GCN | **84.38 ± 0.81** | 97.48 ± 0.06 | 80.00 ± 0.77 | 78.44 ± 0.23 | **91.05 ± 0.15** | 44.26 ± 1.22 | 80.06 ± 0.81 |
| GAT | 83.02 ± 1.21 | 97.00 ± 1.02 | 79.80 ± 0.94 | 77.72 ± 0.71 | 90.38 ± 0.49 | 39.31 ± 2.42 | **81.01 ± 0.23** |
| SAGE | 83.18 ± 0.93 | **97.72 ± 0.70** | 77.42 ± 0.40 | 76.75 ± 1.07 | 90.41 ± 0.10 | 40.22 ± 1.47 | 80.57 ± 0.42 |
| MLP | 58.56 ± 1.75 | 51.74 ± 0.83 | 68.22 ± 0.96 | 70.40 ± 1.17 | 66.43 ± 0.12 | 39.11 ± 1.93 | 72.98 ± 0.49 |
| FAF$_{\text{bestval}}$ | 82.84 ± 0.63 | 90.00 ± 0.39 | **80.96 ± 1.06** | **78.69 ± 0.50** | 78.11 ± 0.38 | **44.59 ± 1.62** | 80.25 ± 0.34 |
| FAF$_4$ | 81.44 ± 0.38 | 90.01 ± 0.51 | 77.20 ± 0.45 | **78.69 ± 0.50** | 78.11 ± 0.38 | 44.02 ± 2.08 | 80.25 ± 0.25 |
| FAF$_{\text{mean,std}}$ | 82.84 ± 0.63 | 90.17 ± 0.51 | **80.96 ± 1.06** | 75.82 ± 1.27 | 77.14 ± 0.52 | 43.83 ± 2.34 | 79.48 ± 0.81 |
| FAF$_{\text{mean}}$ | 82.80 ± 0.70 | 90.00 ± 0.39 | 79.88 ± 0.92 | 76.83 ± 1.19 | 76.36 ± 0.55 | 42.44 ± 1.73 | 79.61 ± 0.56 |
| FAF$_{\text{max,std}}$ | 79.34 ± 0.95 | 88.36 ± 0.74 | 77.52 ± 0.77 | 76.62 ± 0.79 | 75.89 ± 0.30 | **44.59 ± 1.62** | 78.44 ± 0.67 |
| FAF$_{\text{max}}$ | 79.34 ± 0.67 | 86.39 ± 1.22 | 77.18 ± 0.13 | 77.59 ± 1.67 | 75.01 ± 0.43 | 43.03 ± 1.90 | 78.63 ± 0.35 |
| FAF$_{\text{sum}}$ | 81.46 ± 0.62 | 89.96 ± 0.45 | 77.46 ± 0.43 | 76.12 ± 1.08 | 76.90 ± 0.28 | 44.07 ± 1.98 | 76.59 ± 0.36 |
| FAF$_{\text{std}}$ | 79.50 ± 0.39 | 88.93 ± 0.68 | 79.06 ± 1.09 | 73.99 ± 1.67 | 73.80 ± 0.21 | 43.63 ± 1.43 | 76.09 ± 0.26 |

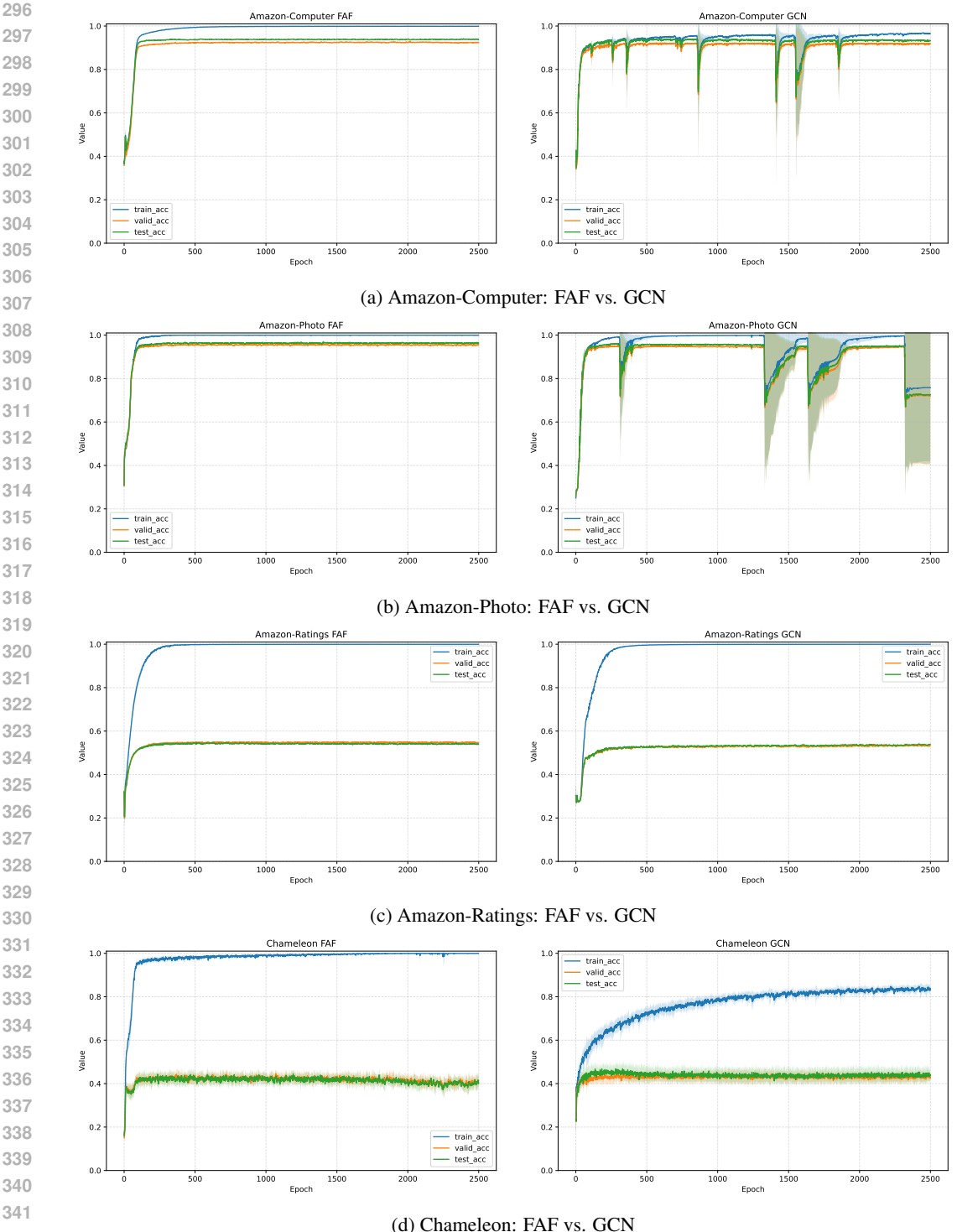

(a) Amazon-Computer: FAF vs. GCN

(b) Amazon-Photo: FAF vs. GCN

(c) Amazon-Ratings: FAF vs. GCN

(d) Chameleon: FAF vs. GCN

Figure 6: Train, validation, and test accuracy of FAF+MLP versus GCN. (i)

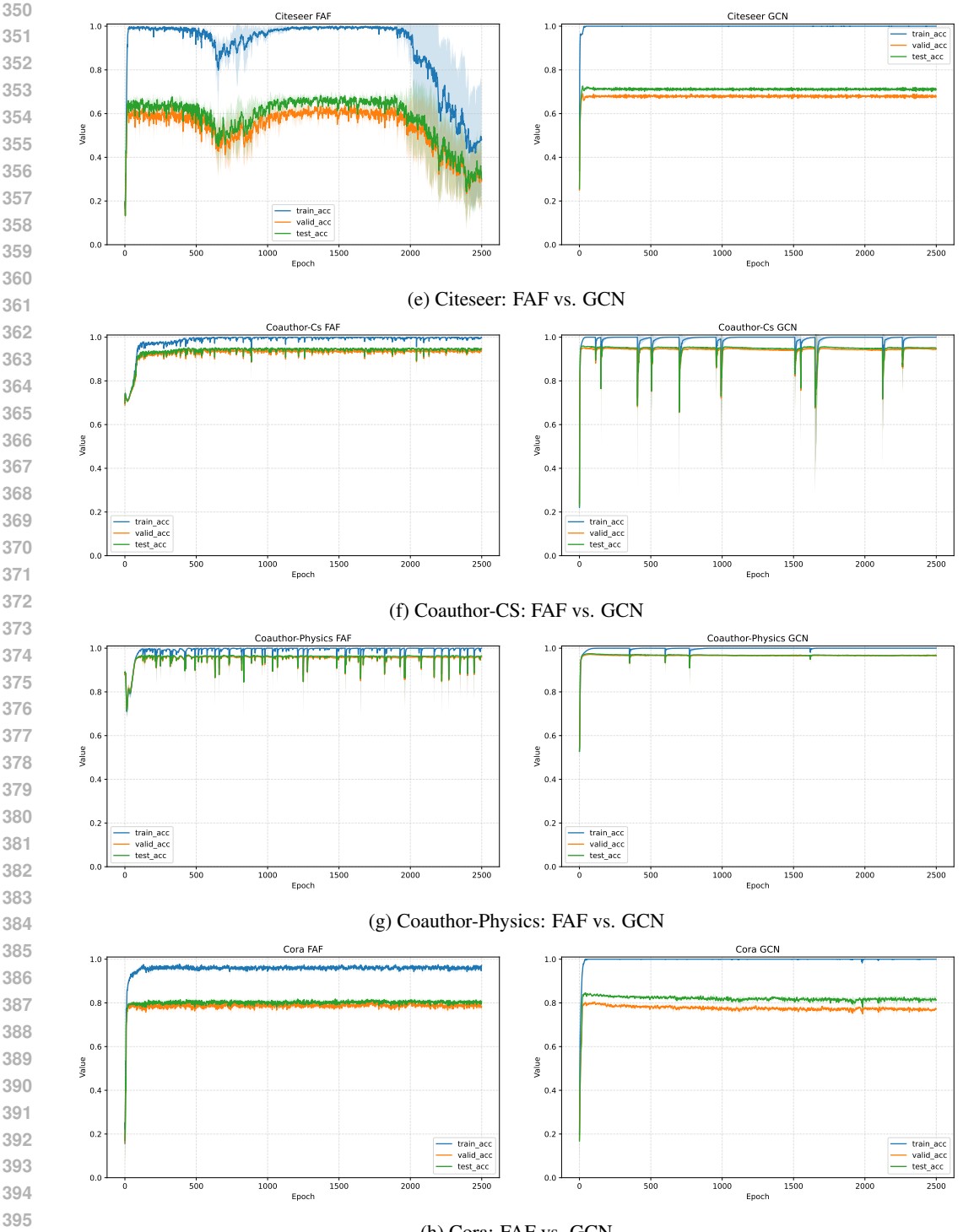

(e) Citeseer: FAF vs. GCN

(f) Coauthor-CS: FAF vs. GCN

(g) Coauthor-Physics: FAF vs. GCN

(h) Cora: FAF vs. GCN

Figure 6: Train, validation, and test accuracy of FAF+MLP versus GCN. (ii)

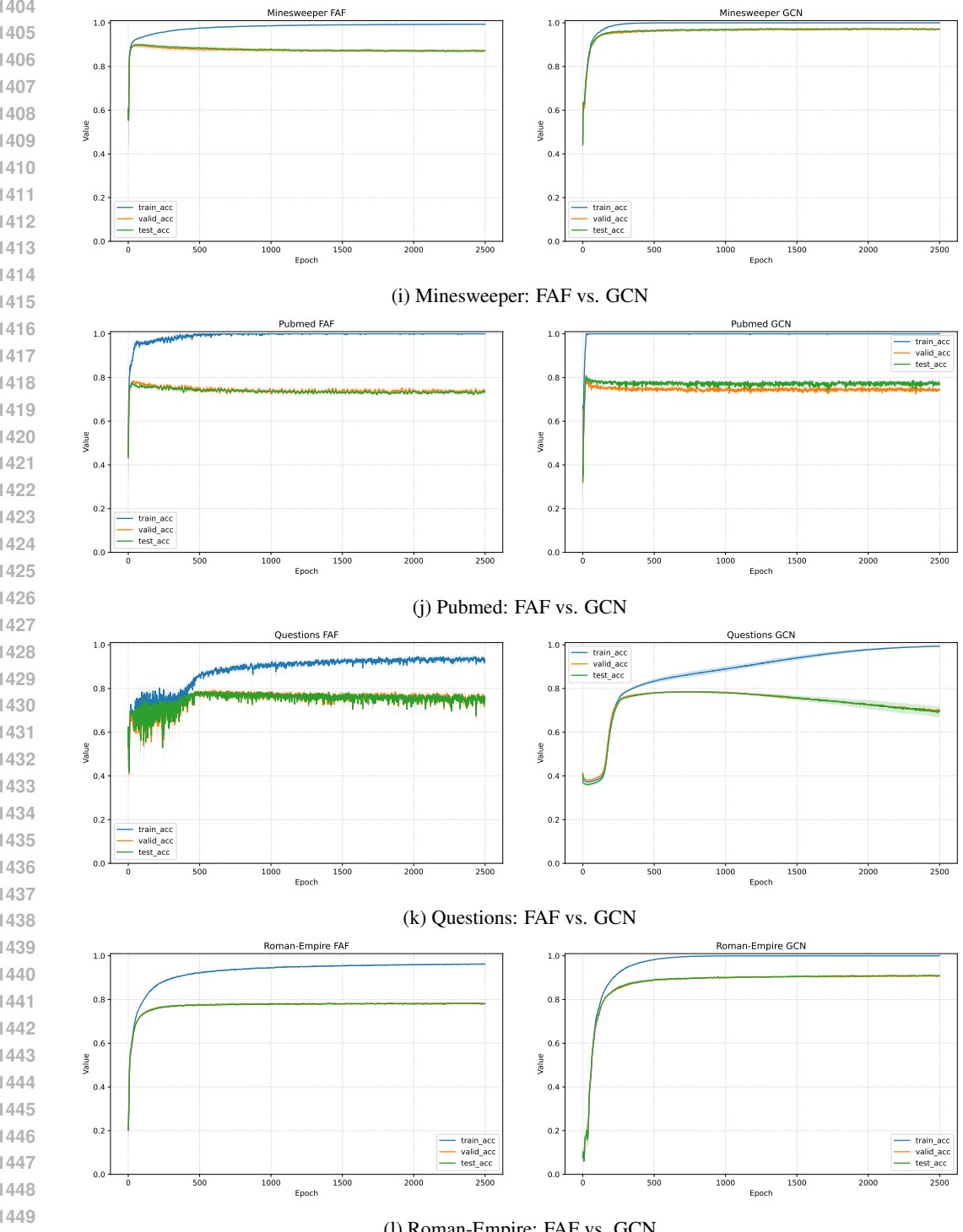

(i) Minesweeper: FAF vs. GCN

(j) Pubmed: FAF vs. GCN

(k) Questions: FAF vs. GCN

(l) Roman-Empire: FAF vs. GCN

Figure 6: Train, validation, and test accuracy of FAF+MLP versus GCN. (iii)

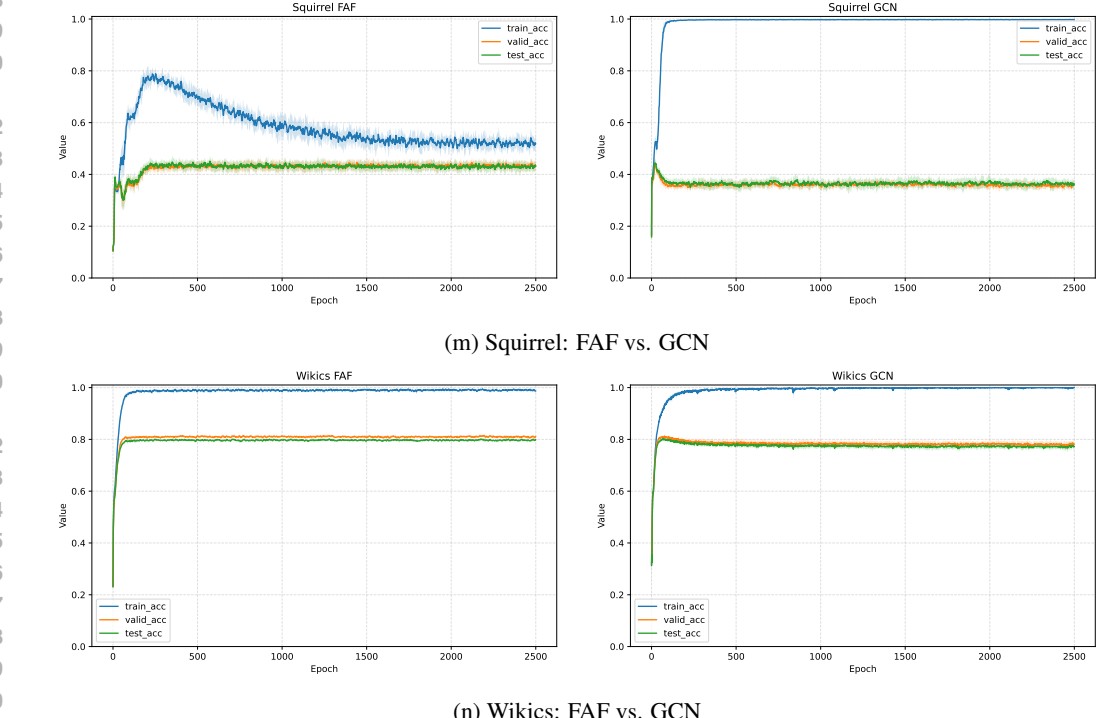

(m) Squirrel: FAF vs. GCN

(n) Wikics: FAF vs. GCN

Figure 6: Train, validation, and test accuracy of FAF+MLP versus GCN. (iv)

# E PRELIMINARY RESULTS ON OTHER BENCHMARKS

We include preliminary results for FAFs on the GraphLand bechmark (Bazhenov et al., 2025) . We do *not* perform the full hyperparameter sweep, therefore we indicate FAFs with an asterisk (*), as there could be better performing versions. We copy baselines from the original paper: MLP, MLP-NFA (one-hop FAFs), GCN and GAT. For FAFs, we only report results for mean+std and mean aggregations. We choose the best validation hyperparameters we have been able to find so far (shown in Table 13) and report the resulting test accuracy in Table 12. While we do not yet achieve the performance of GATs, we approach that of GCNs, and we improve upon MLPs and NFA.

Table 12: Test accuracy of 4 GraphLand datasets (averaged over 10 runs).

|  | artnet-exp | hm-categories | tolokers-2 | pokec-regions |
|---|---|---|---|---|
| ResMLP | 35.07 ± 2.34 | 37.72 ± 0.18 | 41.16 ± 1.13 | 4.88 ± 0.01 |
| ResMLP-NFA | 38.25 ± 0.56 | 48.72 ± 0.38 | 48.14 ± 1.40 | 8.05 ± 0.03 |
| GCN | 43.09 ± 0.38 | 61.70 ± 0.35 | 51.32 ± 0.96 | 34.96 ± 0.38 |
| GAT | 46.62 ± 0.32 | 67.96 ± 0.33 | 53.78 ± 1.34 | 46.17 ± 0.32 |
| FAF*$_{mean,std}$ | 41.56 ± 0.26 | 59.50 ± 0.15 | 52.74 ± 0.53 | 28.44 ± 0.22 |
| FAF*$_{mean}$ | 39.25 ± 0.38 | 54.50 ± 0.13 | 50.72 ± 0.38 | 31.23 ± 0.18 |

Table 13: Best hyperparameters found so far for FAFs on GraphLand datasets.

|  | dropout | lr | bn | hidden channels | weight decay | local layers | mlp layers |
|---|---|---|---|---|---|---|---|
| artnet-exp | 0.7 | 0.01 | 1 | 256 | 0.01 | 3 | 2 |
| hm-categories | 0.5 | 0.001 | 1 | 512 | 0.0005 | 3 | 3 |
| tolokers-2 | 0.3 | 0.0001 | 1 | 512 | 5e-05 | 3 | 5 |
| pokec-regions | 0 | 0.001 | 1 | 512 | 0.001 | 3 | 3 |

# F   COMPARISON TO GRAPH ECHO STATE NETWORKS

Graph Echo State Networks (GESN) (Gallicchio & Micheli, 2010) compute label-independent node embeddings via fixed "reservoir" layers, one per hop, followed by a linear readout. We compare FAFs to this approach in Table 14 as a representative previously proposed simplification of GNNs. We do not include the *coauthor-physics* or *questions* datasets, as GESN exceeds memory capacity on them.

We use the public implementation at `https://github.com/dtortorella/graph-esn`, keeping most GESN-specific hyperparameters as in their example. We set the "depth" (local layers) and "hidden units" (hidden channels) as in the best GCN. Each layer is given 10 minutes to compute its embedding.

As the classifier, we replace the original linear readout with a MLP of the same architecture as for FAFs to provide a fair comparison of the role of the embeddings. For the MLP hyperparameters, we try both the best GCN and the best FAF configurations. Since our different FAFs could share MLP hyperparameters, we expected them to transfer well here, but in this case the GCN hyperparameters perform slightly better. As shown below, FAFs seem to be more suitable for the tested benchmark tasks.

Table 14: Test accuracy of GESN (Gallicchio & Micheli, 2010) against FAFs.

| Dataset | computer | photo | ratings | chameleon | citeseer | coauthor-cs |
|---|---|---|---|---|---|---|
| FAF$_{bestval}$ | 94.01 ± 0.21 | 96.54 ± 0.13 | 55.09 ± 0.24 | 42.96 ± 2.45 | 70.48 ± 1.24 | 95.37 ± 0.17 |
| GESN+MLP | 90.80 ± 0.10 | 92.72 ± 0.27 | 50.34 ± 0.26 | 41.64 ± 3.63 | 41.44 ± 0.34 | 89.99 ± 0.09 |

| Dataset | cora | minesweeper | pubmed | roman-empire | squirrel | wikics |
|---|---|---|---|---|---|---|
| FAF$_{bestval}$ | 82.84 ± 0.63 | 90.00 ± 0.39 | 80.96 ± 1.06 | 78.11 ± 0.38 | 44.59 ± 1.62 | 80.25 ± 0.34 |
| GESN+MLP | 65.78 ± 0.26 | 50.93 ± 1.28 | 64.98 ± 1.57 | 11.76 ± 0.38 | 36.58 ± 1.18 | 73.98 ± 0.85 |

