# OpenReview forum: "Fixed Aggregation Features Can Rival GNNs"
_ICLR.cc/2026/Conference — Submitted to ICLR 2026_

### Official Review · Reviewer_78XL · 2025-10-16

**Soundness:** 2
**Presentation:** 3
**Contribution:** 2
**Rating:** 2
**Confidence:** 4

**Summary:**

This paper presents arguments in favor of using feature-engineering based on multiple, fixed node feature aggregations, rather than having a GNN do the heavy lifting every time. It Is argued that working with tabular ML methods on pre-processed node features offers more interpretability and better training stability than using GNN and that this is supported by empirical experiments on 14 datasets. In addition, the authors introduce a theoretical contribution based on the KAT to motivate the particular feature extraction strategy.

**Strengths:**

Challenging the status quo is always a good idea. This paper does an extremely well job at summarizing prior work that has explored ways of making GNNs simpler, untrained, or heterophily-specific, and then it proposes a solution that seems inspired by multi-aggregation techniques like PNA and concatenation of features like jumping-knowledge networks (JK-Net, Xu et al.). I would like to see more papers devoting so much attention to prior work before proposing a contribution. In this sense, I believe that motivating the FAF scheme by revisiting the KAT under the lens of neighborhood aggregation is an extremely interesting and novel direction. The message-passing architecture is not easy to control and simpler but still effective solutions may even help us reason about different architectural schemes that are more effective than what we use today.

**Weaknesses:**

There are several aspects in the paper that in my opinion require improvements before being accepted at an A* conference.

I think the introduction might benefit from stronger justifications when motivating the proposed technique. For instance, the second paragraph does not seem to necessarily raise the basic question in line 42, rather it could motivate any architectural study of GNNs as a whole. It would be better to find a stronger argumentation that is specific to the aggregation.

The introduction is also quite difficult to read due to some ambiguities. It is often unclear whether the authors mean “graph convolution” or simply the “aggregation operator” when referring to “learning the aggregation”. A statement like “we find that untrained aggregators yield useful features” is true for both GNNs and FAF if one refers to the permutation-invariant operator over neighbors, so I would like to suggest that the authors take more care in disambiguating these cases in the text.

FAF reminds me of an untrained PNA followed by an MLP classifier. I would like to disagree, however, with the statement that FAF can bring more interpretability than current methods that provide an interpretation in the form of a subgraph. The example in Section 3.1 is very ad-hoc, and it does not transfer to a more convoluted scenario where features are continuous and their mixing with different aggregators could make no sense. The experiments in the paper in this direction are not convincing to me, unfortunately.

Section 4 presents interesting theoretical results, which I could not check in detail due to the reviewing workload amidst my other commitments. Apologies for that. I think it is a fresh perspective of neighborhood that I had not heard of. At the same time, I believe there is a big gap between the theoretical result and the empirical experiments, because of the use of non-injective functions invalidates all the theoretical arguments. Please let me know if I misunderstood something here.

In this sense, the answer to the question of Section 4’s title seems to be a sound “yes”: we need to learn aggregators because we have no way of learning injective functions by repeated application of known fixed aggregators. The theoretical arguments, in my opinion, would make for a very nice paper if analyzed and expanded further, and the same could be said for the FAF method. I understand the reasoning behind the current paper’s structure, but in my opinion the paper does not do good justice to neither contribution, since FAF takes too big of a step compared to the theoretical arguments, which are missing some more qualitative analysis.

I have seen a very interesting reference to GESN and following works. I am aware of that research line: GESN variants have achieved very good results on at least subset of these datasets. Being correctly mentioned as related work, in the sense that the untrained reservoir of GESN acts as an “alternative FAF”, though not supported by theory, I am wondering why the authors did not compare empirically with this family of methods.

The most concerning issue on my end remains the empirical evaluation. When one introduces a new class of models, parametrized by a set of hyper-parameters, the usual way of evaluating the empirical risk is by running a model selection, selecting the best hyper-parameters on a validation set, and then evaluating the best configuration (assuming only one in a hold-out data split) on the test set. What I see in Table 1 is instead an instance of hyper-parameter tuning of the hyperparameter R (the set of aggregators to use) on the **test set**. There should be a single line for the class of FAF models in each of these tables, representing the fact that the set R was tuned separately on the validation set for each dataset together with the other hyper-parameters. This is a grave mistake in my understanding. Similarly, all ablation analyses should refer to the validation set performance without looking at the test set, but it does not seem to be the case.

Overall, I really like the perspective of the authors and I encourage them to revise the paper, possibly following some of the suggestions in this review. At the same time, I do not feel I can recommend acceptance of the paper in its current shape.

**Questions:**

Questions:
- I would like to ask the authors where in the paper they supported their statement of line 86 that learnability and numeric stability govern practical success in addition to expressiveness, and why the authors think that this was not clear in the past.
- Can you easily extent your theoretical arguments to F>1, or are they limited to F=1?
- The first two suggestions for future work (lines 358-9) look a bit underdeveloped and generic with respect to the authors’ contribution. Could you elaborate a bit better why they are relevant in this context?
- Potential typo: $\mathcal{X}$ may be undefined in the paper.

---

> ### Author Response · Authors · 2025-11-20
> **Response to Reviewer 78XL**
>
> We thank the reviewer for the thorough review, for appreciating our extensive treatment of prior work, and for describing the KA-based perspective as extremely interesting and novel. We address their concerns in the following points.
>
> ### Weaknesses
> 1. Leading sentence to Paragraph 3
>
> By narrowing the performance gap between classic GNNs and more complex models, the results by Luo et al. (2024) naturally prompt the question which components of the graph-learning architectures are truly necessary for strong performance, especially when additional complexities appear to offer limited gains. This, in turn, leads to another question: how much more can we remove without sacrificing accuracy? In the case of GCNs, what remains are the learnable weights in the graph convolution, so it is natural to scrutinize the importance of this component as well. The results of Luo et al. (2024) clearly demonstrate the viability of simplifying graph models, and our work pushes this line of inquiry further. We are happy to add a sentence in the revision at the beginning of the paragraph to make this connection more explicit.
>
> 2. Graph convolution vs. aggregation operator
>
> We have two distinct concepts that we consistently refer to in the introduction: “aggregators” and “aggregation”. “Aggregators” are the fixed operations like sum, mean, the KA function, etc. We informally defined them by the examples in Line 52. Note that we also use the word “reducers”. “Aggregation”, on the other hand, is any process in which the neighborhood of a node gets aggregated. This includes using fixed aggregators, but also a convolution layer, an attention layer, reservoirs, etc. They can be parameterized or not, and learnable or not. In our paper, we discuss the viability of exclusively using “aggregator operators” on their own for “the aggregation process”, a concept we call “fixed aggregation”. While this is theoretically unambiguous, we agree that the two words are very similar to each other for readers, so in the revision, we have changed the occurrences of the word “aggregator” by the word “reducer” or the phrase “(fixed) aggregation function” across the paper, and we explicitly mention their exchangeability in the revised introduction.
>
> 3. Interpretability
>
> Our aim is not to claim that FAF yields universal or complete interpretability, nor that simple feature-level tools are always preferable to subgraph-based explanations. Rather, what our results on standard node-classification benchmarks suggest is that fixed, shallow aggregations already suffice for matching strong GNN performance. In this regime, tabular explanations (e.g., which hop, which reducer, which original feature matters) are often enough to understand what the model actually uses, and more elaborate graph-specific interpretability machinery may not be warranted by the underlying data. We fully agree that on richer datasets, where genuinely complex message passing is necessary, more sophisticated interpretability methods are needed. Our contribution is to show that for widely used benchmarks, the bar for “necessary complexity” in both architectures and explanations appears much lower than commonly assumed.
>
> 4. Use of non-injective functions in the experiments
>
> On the one hand, we show that injective aggregation functions exist. Therefore, learning the aggregation is not required for theoretical expressivity. However, there seem to be extra factors that currently make our injective aggregator empirically ineffective on most datasets. Note that we do obtain improvements on Roman Empire, suggesting that for the other datasets the issue is likely based on numerical error and instability. On the other hand, our experiments with non-injective functions show that most current benchmarks do not require injectivity to reach comparable performance, and that the relevant distributional information is easily learned by the MLP. This opens up new directions: designing fixed, injective, but more distributional aggregators, or reducing the number of learnable parameters in general, rather than focusing solely on theoretical injectivity. These are precisely the points developed in Subsection 4.3, and especially in the final insight box of Section 4, which we consider central to the paper.
>
> 5. Relating our theory to our experiments
>
> To tighten the theory-experiment link in the revision, we have expanded Section 4.3 to connect the hypotheses more directly to the FAF empirical results, which serve as our main evidence in support of these hypotheses. We have also extended the discussion of possible future directions that our results could inspire.

---

> > ### Author Response · Authors · 2025-11-20
> > **Response to Reviewer 78XL (cont.)**
> >
> > 6. GESN
> >
> > In our experimental setup, we include an ablation comparing our usage of a well-tuned MLP as the classifier improving upon only a linear layer, as is common in methods such as GESN. In principle, we cannot exclude the possibility of a remarkably strong fixed feature representation that would render a simple linear classifier sufficient. In practice, the extensive hyperparameter tuning required for the MLP indicates that having a more powerful classifier is often highly effective.
> >
> > As for a direct comparison, we obtained the PyTorch GESN repository (dtortorella/graph-esn, from the first author of Dynamic GESN) and tried running their Cora example. Unfortunately, the StaticGraphReservoir embedding creation does not converge OOTB with the given hyperparameters (given 1h on GPU A100 for each layer). After that time (1h), we stop iterating and obtain the current state of embedding. This does not seem to give good accuracy (Train: 90.71, Test: 41.20, Valid: 37.80). We assume that this is likely an implementation malfunction. Furthermore, neither can we assume that the node-classification results reported in the respective papers use the same splits or number of runs as ours (as the baselines seem quite different), so copying their numbers would not be fair. Nor can we perform adequate hyperparameter tuning of GESN, as there is no easily findable indication of reasonable hyperparameter ranges or their interactions.
> >
> > 7. Empirical evaluation
> >
> > First, the reported values are indeed validation-set performance obtained without looking at the test set, as required by the reviewer and stated in the first line of Section 5. To verify that this does not harm generalization, we refer to Appendix D, Fig. 6, where we plot the training, validation, and test curves of our preferred method FAF4 and GCN. Test and validation closely match in all cases. Usually, the test accuracy is higher. As also discussed in Appendix D, we mainly report validation performance because our focus is model selection, but we show that the test performance follows exactly the same insights. We are, of course, happy to also include the corresponding test-accuracy table in D.1, which (in line with Fig. 6) supports our insights: On all datasets, FAFs achieve a performance within ±1% of the best classic GNN, except for those already mentioned in the main text (Citeseer, Roman-Empire, and Minesweeper), and Cora, which shows an unusually large gap between test and validation accuracy for GCNs (see Fig. 6.h).
> >
> > Second, we emphasize that our preferred model is FAF4, and that only this version is hyperparameter-tuned. All other FAF variants are treated as ablations and inherit the same hyperparameters, as mentioned in Section 5.2. We report the main results table only for space reasons and to avoid duplicating the classical baselines. Note also that the ablations in Table 1 correspond essentially to feature-selection subsets of FAF4, which indicates that a more elaborate classifier with better feature-selection capabilities could yield further improvements and that the information required for the task is already contained in the fixed aggregation features. We have added a sentence in the revised Section 5.2 to make this explicit.
> >
> > Third, note that the comparison line in Table 1 separates all classical models and all FAF variants. Accordingly, the choice of classical model can also be viewed as a “parameter,” and arguably a worse one, since each of them is tuned on the validation set. Yet, test sets show the same trends. Nevertheless, to avoid distracting from the main message of the paper, we agree to revise the improvement line so that it compares all classical models only to FAF4. This change does not affect the conclusions of our empirical findings.
> >
> > ### Questions
> >
> > 1. Learnability and numerical stability over expressiveness
> >
> > In the paper, we show that our proposed injective aggregator is expressive enough to represent any neighborhood function, but that it is difficult to learn in practice, as evidenced by Table 8 and discussed in the first paragraph of Section 4.3. Moreover, the fact that MLPs require extensive hyperparameter tuning indicates that, while they can model the problem, their learnability is also challenging. This highlights a clear gap between studying expressivity through existence results and actually designing architectures that are learnable in practice. Many works proposing new architectures prove that they are sufficiently expressive, but pay less attention to how these changes affect trainability or learnability—an aspect we argue is equally important and still underexplored.
> >
> > 2. Theoretical argument for more than one feature
> >
> > Our theory applies to more than one feature, i.e. $F \geq 1$. We have corrected that typo.

---

> > > ### Author Response · Authors · 2025-11-20
> > > **Response to Reviewer 78XL (cont.)**
> > >
> > > 3. Our suggestions for future work
> > >
> > > We have extended the discussion of future work in the main text in Section 4.3, which was previously limited due to space constraints:
> > >
> > > (1) Feature/reducer engineering: FAFs highlight untapped potential for designing meaningful node features that encode graph structure, require less learning, potentially preserve more/but ideally only relevant information, and allow for higher learning efficiency. In combination with partial feature learning, they might form the basis of a new generation of graph based learning architectures.
> > >
> > > (2) Moving beyond injectivity: As our theory and empirical results highlight, improving GNN expressiveness and thus injectivity alone is not likely to inspire practical improvements on current benchmarks, as those can be competitively solved even with simple, non-injective aggregation. We therefore call for a shift in focus from mere injectivity to other learning properties—a theoretical gap to be addressed not only for FAFs but for GNNs in general.
> > >
> > > (3) Benchmarks: Enough information to solve current benchmark tasks is already contained in early hops and can be extracted with simple, non-injective aggregation. If we really want to showcase the capabilities of GNNs to learn meaningful features, we need more difficult benchmarks that require this ability.
> > >
> > > 4. Typo of $\mathcal{X}$
> > >
> > > We thank the reviewer for pointing out this typo. We changed the first line of section 4 from $X$ to $\mathcal{X}$.

---

> ### Comment · Reviewer_78XL · 2025-11-21
> **Response to Authors**
>
> The authors did add a simple connecting statement in the introduction, which is not exactly what I meant, but I appreciate the effort. Also, thank you for taking care of the ambiguities in the introduction: the use of similar words was making the understanding difficult if one is not extremely careful about the semantics while reading.
>
> About interpretability: it is clear that there exists no universal/complete interpretability, but arguing that shallow aggregations already suffice performance-wise is different from arguing they are interpretable. The tabular explanations still seem not very interpretable, considering, as I already mentioned, that the example of Figure 2 corresponded to a toy use case. I am not sure if it is a good idea to talk about interpretability in this paper.
>
> I do remember that the authors already considered that learning injective functions is hard and may not be that helpful in practice: it was not my intention to use a considered limitation as a reason for weakness, my point was that there is a clear disconnect between theory and practice in this paper. I had a look at the revised version, where the authors suggest to move beyond injectivity since it is not likely to bring practical improvements. This makes me wonder again what is the true contribution of the paper: if the authors cannot show that their learned aggregation functions is injective, how can they claim that non-injective ones reach comparable performances? At this point, proposing to move past injectivity feels like giving up on the idea because it does not work. My concerns remain in this sense: the paper's narrative, contribution and impact might greatly improve if it focused on analyzing injectivity rather than comparing with the state of the art. I understand that many reviewers ask for empirical comparisons, but these conferences also allow for infinite appendices, so I think the authors could still focus on what, in my personal opinion, matters more for the main paper.
>
> About GESN: I was not referring about the comparison between linear and MLP, but rather to the comparison with the untrained embeddings computed by the graph echo state network framework. Thank you for trying out the Dynamic GESN method, and it is unfortunate that you could not make it work in this short time span (there is still time, if the authors want). At the same time, the authors would agree I cannot consider this a valid answer to the point, since GESN is a seriously related work, at least empirically, to FAF and as such it should have been included in the evaluation (Disclaimer: I am not an author of GESN-related papers but I find the approach fascinating, though requiring a heavy hyper-parameter tuning stage to work properly).
>
>
> About experiments: I apologize for not spotting the "validation" mention in line 370 and the discussion Appendix D, I will therefore raise this score accordingly since this is not a problem. However, please take further care in making this **very** clear: the choice of the authors to not include test results is so unorthodox that it risks being misunderstood by readers, especially considering the different numbers. In Table 10, the authors should however consider only FAF4 rather than all the other versions in my opinion, as the risk is generally assessed on a single family of models.

---

> > ### Author Response · Authors · 2025-11-22
> >
> > Thank you for the quick response and your continued engagement in the review process. Your feedback really helps us to improve the paper, avoid disambiguities, and carve out the implications of our findings.
> >
> > **Introduction:**
> > We would be happy to adapt the introduction according to what you mean. However, to do so, we believe we have to lay out the main premise and storyline of the paper, as we seem to disagree on the implications of our findings and, as a consequence, how we should build the introduction.
> >
> > **Storyline:**
> > How we see our contributions and the relationships to interpretability and injectivity (these points are all connected):
> >
> > What we find striking is that common GNN layers do not need training. Simple instances, i.e., untrained aggregations (like mean or sum aggregation) suffice. The natural question to ask based on this observation is: How and why can this work? And the answer is not that sum or mean aggregations are injective. Information is lost after the second layer, but this seems to be enough. This is also in line with the fact that (trainable) GNNs tend to require only a few layers to work well. Problems like oversmoothing further limit the depth (in line with the information loss resulting from sum and mean aggregation) unless we use skip connections and concatenation that essentially preserve information from the first layers. Beyond this insight that information loss does not seem to be a big issue for current bechmarks, means and sums even seem to have an inductive bias that helps with the learning tasks. Otherwise, they would not perform so much better than Kolmogorov-Arnold features that actually are injective (so information is not lost), but do not provide representations that are amenable to even MLP classifiers.
> >
> > So what implications does this analysis have for the GNN community? First of all, there might be potential for hand-crafting features that are better than simple neighborhood mean aggregation, because they preserve more information. However, considering that most GNNs do not learn something that performs much better on the considered benchmarks, we need more interesting benchmarks to figure this out.
> > Given that we now have FAFs, we can also make use of the fact that we turned graph based learning problems into tabular ones. One of the advantages that we see is that this data structure has been studied for a long time and the resulting developments transfer now also to graph based learning problems. Interpretability methods for tabular data are just one example that is not central to our line of arguments. This does not diminish the utility of interpretability methods that have been specifically designed for graphs. It just means that we have a larger toolbox available to study the problem from different angles.
> >
> > We are aware that you might disagree and would therefore like to learn more about your opinion and welcome the exchange. We are open to change the emphasis of specific points and contributions, but we would like to emphasize that we consider our contributions to be novel, surprising, and relevant to the GNN community, and that the storyline is a stylistic choice and not a weakness.
> >
> > **Interpretability:**
> > FAFs bring common benchmarks in tabular format, which gives us novel insights into typical graph based learning tasks, which is not obvious from studying GNN architectures. (Interpretability) methods for tabular datasets now transfer to GNN datasets. This does not imply that these methods are strictly more interpretable than custom GNN interpretability tools nor does they make such custom tools obsolete. Yet, they allow us to study the tasks from another angle, which can be quite helpful for interpretation.
> > Furthermore, we do argue that simple neighborhood means or sums are potentially better interpretable than nonlinear transformations, in particular, since they preserve the meaning of the original features.

---

> > > ### Author Response · Authors · 2025-11-22
> > >
> > > **Injectivity:**
> > > We do not advocate moving “past” injectivity (moving without it and thus leaving it behind), but “beyond” it (moving with it and more), meaning that we should be more careful in attributing performance gains to an increase in injectivity, and that the trainability of new architectural elements is extremely important and should receive more focus, for instance. We also do not claim that injectivity cannot induce practical improvements in general. Instead, we unveil gaps between the promised improvements of injectivity and the practical improvements of good optimization, good feature engineering, and a strong classifier. Furthermore, the reason why the non-injective reducers reach comparable performance is that the tasks we usually evaluate on do not require it. We also consider this finding a striking empirical contribution in itself, with the purpose of our work serving as a critical evaluation of the most common benchmarks. Also note that the sentence mentions “improving [..] injectivity alone is not likely to inspire practical improvements *on current benchmarks*”, as they seem to require other ingredients not considered beforehand, and we assert they need to be addressed more thoroughly. We respectfully disagree that further developing injective reducers in this paper matters more than the presented contribution, such as the proposal of a strong new baseline, the critical view of current benchmarks, the extensive empirical evaluation that yields surprising findings, the new angle on graph aggregation, and as mentioned above, the new insights on improvements originating from factors other than injectivity. We consider the design of aggregations that walk the line between injectivity and good feature representation to be a great new avenue that we hope can inspire the community to build both better graph architectures and more difficult benchmarks, but it is out of scope for our current work.
> > >
> > > **GESN:**
> > > We are curious about this suggestion and would really want to compare our results with this method. We have tried to construct the untrained embeddings computed by the graph echo state network framework, but as it is, we cannot provide a fair comparison with them for the explained reasons. Without a codebase that converges and without a guide on their hyperparameters, we can only run it as is and show the non-converged experiments. We would be grateful for a pointer to an official code version of this method with a working dataset and correct hyperparameters out of the box. Please let us know if this is available somewhere.
> > >
> > > Moreover, while this is a very interesting approach, we respectfully disagree that it forms a highly related baseline. The purpose of our empirical investigations is to show that untrained GNN instances define meaningful feature embeddings (despite not being injective). It is an interesting research question for follow-up work to investigate alternative ways to handcraft features. To highlight this, we already cite the suggested approach, discuss it as a precursor of simplifying graph learning paradigms, and have tried to replicate the experiments transparently.
> > >
> > > Also, without this comparison: All our contributions and new insights hold and are as surprising, novel, and necessary to disseminate regardless of the suitability of GESN for the graph learning tasks. Moreover, the fact that it requires heavy hyperparameter tuning, and that this tuning is not standard compared to other methods (that is, many parameters that depend on their specific architectural choices) makes this method also more demanding than our simple proposal to use unparameterized aggregation functions like mean and sum, which as we show, empirically work as well as trained GNNs.
> > >
> > > **Experiments:**
> > > We noticed we still had extra space to add both tables in the main text. We have taken the best validation FAF for the test table, and we retain the full test results in the appendix. We have changed two sentences in the experimental section to account for this change (as the insights are exactly the same), and we had to delete the delta improvements row for space. We thank the reviewer for improving the paper’s portrayal of our experimental results, and we hope that we now have the most satisfactory and clear format for readers to examine our contributions.

---

> ### Comment · Reviewer_78XL · 2025-11-24
> **Thank you for the answer**
>
> I want to thank the authors for the detailed clarifications. At this point I would agree to disagree, because based on some of the points raised we could further develop a discussion but there is limited time to handle all responses.
>
> In particular, I just want to mention:
> - I do not have a good solution for the introduction, because I still believe this paper could be split into two very valuable contributions despite the above clarifications, and therefore the narrative would need to change drastically according to that
> - I am not totally convinced by argument on GESN. It the purpose of the investigation were to show that untrained GNN instances define meaningful embeddings, one might then argue that prior work already did, e.g., GESN, and that your work is not so novel in that respect. In addition, I would expect that prior work is properly compared with, not just cited. Practically speaking, one may want to use the easiest and strongest baseline possible rather than trying all of them all the time. So, if you propose a method that is related in spirit to another, I would have expected a comparison. The alternative, as I mentioned, would have been to focus on theoretical results first and then use such results to propose FAF in a second contribution. I have the impression that the authors should funnel down a bit the scope of their analyses to avoid such criticism. Especially considering that I typically run proper hyper-parameter tuning regardless of the GNN I use; what is "standard" in the community is not necessarily the best way to compute risk estimates
>
>
> To be clear, I will not oppose to a potential acceptance as my score suggests. I will make this clear in the discussion with the other reviewers, but I think this paper holds much more potential than what the current presentation delivers.

---

> ### Author Response · Authors · 2025-11-26
>
> We sincerely thank the reviewer for their continued engagement with our work and are happy to clarify any remaining concerns as far as the discussion period allows.
>
> **Separation of contributions:** We respectfully disagree that our theoretical and empirical contributions could be disentangled. They are tightly coupled: each is necessary to fully understand the other. On the one hand, our theoretical analysis is required to explain the empirical performance we observe, and to motivate and justify the viability of the FAF method. On the other hand, our experiments are intrinsically necessary to convince (and, we believe, to surprise) the reader about the importance and implications of our theoretical discussion, and thus cannot be relegated to a separate contribution. In particular, they reveal that (i) learning aggregations is not a requirement for good performance, as common benchmarks *can* be solved with fixed features and MLPs; (ii) the simplest possible aggregations already suffice, having the right inductive bias (capturing distributional information) to solve these tasks as well as GNNs; and (iii) injectivity is not required to do so, despite being attributed much of the performance gains in the literature. The surprising fact is not that reducers like mean are non-injective, but that such simple, non-trainable aggregations perform so well on their own on these datasets. Thus, our claim is not that FAFs are the best possible aggregation scheme yet, but that they are already sufficient to be competitive with GNNs on these benchmarks, despite the crucial advantage GNNs enjoy in learning task-specific feature representations. This shows that GNNs do not require trainable aggregations to reach current benchmark performance, and that mean/max/etc. provide a surprisingly good inductive bias even though they are not information preserving.
>
> In summary, we argue the following: the easiest possible fixed aggregations are already sufficient for the current datasets, and we also show why fixed aggregations can, in principle, be theoretically sufficient more generally. As we argue in the paper, empirically bridging the two statements is currently impossible because we lack benchmarks that clearly distinguish tasks requiring injectivity from tasks that only require what our simplest aggregations can provide. Note that this insight is only available after having developed *both* our theoretical analysis and our experiments.
>
> This is also the reason why we do not expect many other feature constructions to yield substantial gains over FAFs on these benchmarks, as the simple FAFs already rival GNNs—note also that Luo et al. (2024) compare these GNNs in our same setting to more complex architectures such as graph transformers. This is likewise why we have argued that we do not expect GESN to create a significant performance gap, and why our discussions are orthogonal—or even prior—to improving the construction of hand-crafted or label-independent features. We believe, however, that there is potential for such future work, provided that we have meaningful benchmarks on which genuine progress can be measured.
>
> **Tuning:** Finally, we note that all baselines and our MLPs are carefully hyperparameter-tuned, and we do not mean to suggest that GESN should not be. Our point is that our aggregation is simpler and computationally more efficient, yet still *sufficient* to rival GNNs on these tasks, so our insights apply regardless.

---

> ### Author Response · Authors · 2025-11-26
>
> **GESN:** For completeness, we include results on GESN. We do not include the coauthor-physics or questions datasets, as GESN exceeds memory capacity on them. We set the "depth" (local layers) and "hidden units" (hidden channels) as in the best GCN. Each layer is given 10 minutes to compute its embedding. As the classifier, we replace the original linear readout with a MLP of the same architecture as for FAFs to provide a fair comparison of the role of the embeddings. As shown below, FAFs seem to be more suitable for the tested benchmark tasks. We would be happy to add these results to the revised Appendix F.
>
> Table 1: Test accuracy of GESN against FAFs.
>
> | Method        | computer     | photo        | ratings      | chameleon    | citeseer     | coauthor-cs  |
> |--------------|-------------:|-------------:|-------------:|-------------:|-------------:|-------------:|
> | FAF (best val) | 94.01 ± 0.21 | 96.54 ± 0.13 | 55.09 ± 0.24 | 42.96 ± 2.45 | 70.48 ± 1.24 | 95.37 ± 0.17 |
> | GESN+MLP     | 90.80 ± 0.10 | 92.72 ± 0.27 | 50.34 ± 0.26 | 41.64 ± 3.63 | 41.44 ± 0.34 | 89.99 ± 0.09 |
>
> | Method        | cora        | minesweeper  | pubmed       | roman-empire | squirrel     | wikics       |
> |--------------|------------:|-------------:|-------------:|-------------:|-------------:|-------------:|
> | FAF (best val) | 82.84 ± 0.63 | 90.00 ± 0.39 | 80.96 ± 1.06 | 78.11 ± 0.38 | 44.59 ± 1.62 | 80.25 ± 0.34 |
> | GESN+MLP     | 65.78 ± 0.26 | 50.93 ± 1.28 | 64.98 ± 1.57 | 11.76 ± 0.38 | 36.58 ± 1.18 | 73.98 ± 0.85 |

---

### Official Review · Reviewer_fmA9 · 2025-10-16

**Soundness:** 3
**Presentation:** 2
**Contribution:** 3
**Rating:** 6
**Confidence:** 5

**Summary:**

The paper shows that a combination of non-learnable feature aggregation a an MLP can provide performance competitive with GNNs on many classic datasets for node classification. First, the paper proposes concatenating features obtained from neighborhood aggregations with different radius with the original node features to essentially convert a node classification task to a tabular classification task. Then, the paper investigates some theoretical aspects of such aggregations, showing that it is possible to design aggregations that preserve all neighborhood information (but, as the paper admits, such aggregations are not very practical). Then the paper conducts experiments with simpler aggregations and shows that MLPs on top of such aggregations can often rival GNNs.

**Strengths:**

- The main idea is simple but useful, its empirical performance can sometimes be quite impressive.
- Experiments are conducted on a vast range of datasets, strong baseline models with adequate hyperparameter search spaces are used.
- The paper raises timely questions regarding the adequacy of standard node classification benchmarks for the evaluation of complex models.

**Weaknesses:**

- The interpretability example with the minesweeper dataset (section 3.1) is wrong due to a misunderstanding of what the node features are. As described in [1], where the dataset was proposed, the node features use one-hot encoding for the number of neighboring mines, not binary encoding (see page 7 of [1], the Minesweeper paragraph). Due to this mistake, the explanation that the authors provide for how the model uses the features is entirely wrong. I do not consider this a serious issue, as it is just a minor example that does not affect the main points of the paper, but it needs to be fixed.

- The theoretical contributions seem not particularly interesting. First, Section 4.1 relies on the assumption of feature orthogonality, which is not very realistic. Even bag-of-words features are not orthogonal (and bag-of-words is an extremely outdated technique in 2025, but that is the problem of standard graph ML benchmarks, not of the current paper), and other feature types are typically even further from being orthogonal. Then, section 4.2 proposes a theoretical construction that, as the authors themselves admit (which is commendable), is not very practical. This leads to the theoretical sections being rather disconnected from the experimental sections. This is partially alleviated by some interesting discussions in Section 4.3, but they are not fleshed out enough in my opinion. I suggest shortening Section 4.1 and giving more space to the discussions in Section 4.3 (perhaps by providing more evidence for the hypotheses) to improve the paper.

- It is a strong point of the paper that it uses a lot of datasets for experiments and evaluates improved and well-tuned GNNs from [2] rather than weaker models that are often used as baselines in other works. However, this raises the question: if the authors use the codebase of [2] and also use almost the same hyperparameter search space, why are the reported results sometimes significantly different than those in [2]? For example, on the cora dataset, the reported result for GCN is 81.28, while [2] reports 85.10 (note that a similar results would make GCN rather than FAF4 the strongest model on cora in the current paper). There are similar discrepancies for some of the other datasets. What is the reason for them?

I am willing to raise my score if my concerns are addressed.



[1] A critical look at the evaluation of GNNs under heterophily: Are we really making progress? (ICLR 2023)

[2] Classic GNNs are strong baselines: Reassessing GNNs for node classification (NeurIPS 2024)

**Questions:**

See weaknesses.

Some other suggestions for paper improvement:

- The paper does not discuss efficiency at all, but it could potentially be another strong point of the FAF approach: precomputing aggregations once and then training MLPs on top of them is much faster than training GNNs (graph aggregation is typically the slowest operation in GNNs). I suggest discussing it and possibly even providing training times.

- The paper mentions a couple times that its results imply that the current benchmarks used in graph machine learning are inadequate for evaluating complex models. This could be discussed more and positioned within the recent line of works that also discuss and/or address this issue. Specifically, [3-5] discuss the problems with current benchmarks ([3] is briefly mentioned in the current work), and [5] proposes better datasets for node property prediction. Note that [5] additionally uses neighborhood feature aggregation (NFA) mechanism which is similar to a 1-hop version of FAF, and also shows that it can be a strong baseline. Even more recently, a concurrent work [6] also proposes better benchmarks for graph machine learning.

[3] Graph learning will lose relevance due to poor benchmarks (ICML 2025)

[4] No Metric to Rule Them All: Toward Principled Evaluations of Graph-Learning Datasets (ICML 2025)

[5] GraphLand: Evaluating Graph Machine Learning Models on Diverse Industrial Data (NeurIPS 2025)

[6] GraphBench: Next-generation graph learning benchmarking

---

> ### Author Response · Authors · 2025-11-20
> **Response to Reviewer fmA9**
>
> We thank the reviewer for their careful reading, for acknowledging the usefulness and empirical results of our work, and for the constructive suggestions. We address their concerns in the following points.
>
> ### Weaknesses
>
> 1. Interpretability example
>
> We thank the reviewer for the correction on the meaning of the features. We now include it in the revision and have changed the resulting qualitative interpretation, accordingly, which still highlights the utility of our approach.
>
> 2. Theoretical contributions
>
> We respectfully disagree that the theoretical component is uninteresting or disconnected from the empirical results. Section 4 uses a novel perspective on KA to show that fixed injective neighborhood aggregations are, in principle, sufficient for expressivity, thereby separating expressivity from the equally important aspects of learnability and numerical stability. This highlights the surprising fact that expressivity alone does not justify learning the aggregation. With it, we explain our empirical finding that on many benchmarks simple, shallow, non-injective statistics already capture most relevant neighborhood information, and it further reveals other interesting insights, such as deeper aggregations suffering from difficult optimization and overfitting. To tighten the theory-experiment link, we have removed a paragraph in Section 4.1 and expanded Section 4.3 of the revision to connect the hypotheses more directly to the FAF empirical results, which serve as our main evidence in support of these hypotheses, as well as extending our descriptions of the three possible future directions that we uncover with our results.
>
> 3. Different metrics than the benchmark
>
> For the baseline GNN methods, we do not perform a full hyperparameter sweep. Instead, we use the best configurations reported in the original work, as provided in the codebase. The baselines are run directly using the authors’ released implementations from their official GitHub repository (LUOyk1999/tunedGNN). Moreover, note that, because our study focuses on model selection, we have reported validation performance in the main text. Test accuracies are usually higher. To verify that we generalise well, we report train, validation, and test learning curves for our main method FAF4 and the GCN baseline in Appendix D Fig.6, confirming that validation and test metrics match throughout training. We are, of course, happy to also include the corresponding test-accuracy table in D.1. In line with Fig. 6, our insights also apply to the test accuracy: On all datasets, FAFs are within ±1% of the best classic GNN, except for those already mentioned in the main text (Citeseer, Roman-Empire, and Minesweeper). Only Cora shows an unusually large gap between test and validation performance for GCNs (see Fig. 6.h).

---

> > ### Author Response · Authors · 2025-11-20
> > **Response to Reviewer fmA9 (cont.)**
> >
> > ### Questions
> >
> > 1. Efficiency and training times
> >
> > We thank the reviewer for this very interesting point. We have added a new paragraph in our list of advantages in Section 3.1, noting that the increase of input features also affects runtime, but it could be solved by feature reduction strategies. We also include training times in Table 4 in the revised appendix.
> >
> > \paragraph{Efficiency.}  Precomputing aggregation once and then training an MLP on top is far more scalable than repeatedly running message-passing layers and backpropagating through them, as required in GNNs. However, as the number of reducers, original features, and hops in FAF increases, so does the input dimensionality, which in turn enlarges the parameter count of the MLP’s first layer. This issue could be mitigated through common feature reduction techniques. For the original features, we report the average training runtimes of our FAF and GNN models in Table~4. FAFs are generally more efficient, particularly when using a single reducer.
> >
> > Table 1: Empirical training time in seconds of FAF and GNN models, averaged over runs.
> >
> > | Dataset          | GCN       | GAT       | SAGE       | FAF_4    | FAF_2    | FAF_1   | MLP       |
> > |------------------|-------------|-------------|--------------|--------------|--------------|-------------|-------------|
> > | computer         | 127.30±0.58 | 29.33±3.22  | 45.33±0.58   | 42.67±0.58   | 25.00±0.00   | 17.00±0.00  | 9.33±0.58   |
> > | photo            | 82.00±0.00  | 26.00±0.00  | 33.33±0.58   | 66.67±0.58   | 36.00±0.00   | 21.33±0.58  | 7.33±0.58   |
> > | ratings          | 140.30±0.58 | 161.30±0.58 | 330.00±0.00  | 46.00±0.00   | 26.00±0.00   | 17.33±0.58  | 10.00±0.00  |
> > | chameleon        | 22.30±0.48  | 15.20±0.42  | 18.00±0.00   | 44.50±0.97   | 22.50±0.53   | 15.10±0.32  | 10.00±0.00  |
> > | citeseer         | 16.20±0.45  | 20.00±0.00  | 26.00±0.00   | 62.80±1.79   | 36.20±0.45   | 25.80±1.79  | 13.60±0.55  |
> > | coauthor-cs      | 157.00±0.00 | 70.33±0.58  | 301.30±20.50 | 296.30±1.16  | 159.00±0.00  | 91.00±0.00  | 26.33±0.58  |
> > | coauthor-physics | 65.33±0.58  | 190.00±0.00 | 400.30±0.58  | 2383.00±0.58 | 1004.00±1.00 | 532.70±0.58 | 183.00±0.00 |
> > | cora             | 16.40±0.89  | 20.20±0.45  | 11.20±0.45   | 46.80±0.45   | 26.00±0.00   | 17.00±0.00  | 8.20±0.45   |
> > | minesweeper      | 69.67±0.58  | 100.30±0.58 | 68.67±2.08   | 19.00±0.00   | 20.00±3.46   | 19.00±0.00  | 18.00±1.73  |
> > | pubmed           | 20.20±0.45  | 42.00±0.00  | 78.20±0.45   | 34.00±0.00   | 19.20±0.45   | 12.00±0.00  | 6.20±0.45   |
> > | questions        | 650.00±0.00 | 258.00±1.00 | 363.70±0.58  | 180.70±0.58  | 122.70±0.58  | 94.00±0.00  | 64.67±1.16  |
> > | roman-empire     | 240.30±0.58 | 294.30±0.58 | 93.00±0.00   | 38.67±2.89   | 23.33±0.58   | 18.00±0.00  | 14.67±2.89  |
> > | squirrel         | 29.00±0.00  | 87.10±0.32  | 24.70±1.89   | 43.30±0.48   | 25.20±0.42   | 18.00±0.00  | 11.70±0.48  |
> > | wikics           | 60.00±3.46  | 97.33±0.58  | 27.33±0.58   | 10.00±0.00   | 7.33±0.58    | 7.00±0.00   | 7.00±0.00   |
> >
> >
> > 2. Related work extension
> >
> > We thank the reviewer for this relevant set of works, and we agree to include a paragraph in the revision’s related work on recent literature that aims to improve GNN benchmarks:
> >
> > Our work also connects to the growing literature on properly benchmarking GNNs and what constitutes a meaningful graph-learning dataset. For graph classification, \citet{Errica2020A} show that, under controlled protocols, simple and even structure-agnostic baselines can rival complex GNNs, suggesting that common benchmarks often fail to exploit graph structure. Recent analyses likewise warn that graph learning risks losing relevance without application-grounded benchmarks \citep{bechler-speicher2025position} and principled criteria for dataset quality beyond accuracy \citep{coupette2025no}. On the dataset side, \citet{bazhenov2025graphland} recently introduced industrial node property prediction benchmarks with graph-agnostic baselines. Their neighborhood feature aggregation (NFA), which augments tabular models with one-hop aggregated neighbor statistics, can be seen as a one-hop instance of our FAF construction. Concurrently, \citet{anonymous2025graphbench} propose a suite of benchmarks to standardize graph-learning evaluation. In this context, our FAF approach serves as a simple stress test of whether proposed graph benchmarks genuinely benefit from learned message passing, and we argue that such well-tuned, fixed, multi-hop baselines should be routinely included when assessing new graph models and datasets.

---

> ### Comment · Reviewer_fmA9 · 2025-11-22
>
> I thank the authors for their detailed response which addressed most of my concerns. I am still not particularly sure that the theoretical results are as important as the authors claim them to be, but my appreciation of these results definitely increased after the authors' replies and paper update. All other aspects of the paper are very good, and I am also impressed by the experimental results on the recently proposed realistic GraphLand datasets provided in a reply to another reviewer (I hope, these results will be included in the final version of the paper, as they showcase the FAF performance on more practically-relevant datasets than those typically used in academic papers). Overall, I believe this is a timely work that will be useful for the graph machine learning community, and I raise my score to 8.

---

> > ### Author Response · Authors · 2025-11-25
> >
> > We sincerely thank you for your encouraging assessment and very helpful feedback. We have added the GraphLand results in the revision as suggested (appendix E).

---

### Official Review · Reviewer_6Pr9 · 2025-10-23

**Soundness:** 4
**Presentation:** 3
**Contribution:** 3
**Rating:** 4
**Confidence:** 4

**Summary:**

This work presents Fixed Aggregation Features, a simple technique that combines neighborhood aggregations of different types (e.g., mean, max, std), applies them to neighbors at different hops, concatenates the obtained information with the original node features and trains an MLP on top of these representations. The experiments show that such a technique can provide results comparable to or even better than standard message passing networks when measured on classic graph ML datasets. Thus, the work introduces a previously overlooked class of strong graph ML baselines and questions the relevance of classic citation and co-authorship networks with bag-of-word node features for evaluating advanced graph ML models.

**Strengths:**

1. An introduction of non-learnable multi-hop feature aggregation as a preprocessing step is very simple, intuitive and practical approach to augment the original node features with graph-based information.

2. A discussion of the problems in optimization procedure of GNNs and theoretical properties of FAF, which also explains why a standard MLP on top of simple graph-based aggregations can perform on par with standard GNNs.

3. An extensive empirical study showing that FAF enables to achieve nearly the same results as GNNs, together with ablation study showing that even a single aggregation type in FAF can be sufficient for some datasets.

4. A number of additional experiments showing that theoretically lossless Kolmogorov-Arnold aggregations are not so effective in practice, while more simple aggregations like mean or max should be preferred for constructing FAF.

**Weaknesses:**

As the main weakness of current version, I see the choice of graph datasets for experiments. There is a recently introduced GraphLand benchmark [1] that provides both classification and regression tasks from industrial applications, includes both homophilous and heterophilous graph datasets, and contains rich heterogeneous tabular node features. Moreover, this work introduces Neighborhood Feature Aggregation (NFA) that seems to be a specific instance of FAF using mean, max and sum aggregations over 1-hop neighborhood. It might be very relevant for this particular study and thus should be discussed as related work.

It is very interesting to see whether the observations about the importance of the closest neighborhood hold and how the performance of simple MLP on top of FAF representations transfers to GraphLand datasets. I admit that this benchmark appeared very recently, but I would highly recommend to include the experiments using at least the RL (random low) data split, as it could help the authors not only investigate their hypothesis regarding the need for more relevant graph datasets, but also significantly strengthen their empirical study in general. If the authors manage to provide such additional results, I am ready to increase my score.

[1] GraphLand: Evaluating Graph Machine Learning Models on Diverse Industrial Data, NeurIPS 2025

**Questions:**

1. A couple of comments regarding the Theorem 1 and its proof in Appendix A.2.

- I am not sure that the term "unique function" in the formulation of this theorem is clear for me. As I understand, "uniqueness" means *preserving the information about elements of the original multiset and making it possible to know how many elements in a multiset have a particular feature value*. If this is true, I would ask the authors to expand the formulation and make it more explicit.

- I also feel that some explaining comments about decomposing any multiset function $f$ in the mentioned form are missing. As I understand, any multiset function $f$ depends on the counters of unique elements observed in it. Since we preserve the whole information about multiset under the transformation $\Phi$, we can restore these counters by using $\Phi^{-1}$ and then apply the desired $f$ to them. If the reasoning is correct, I would ask the authors to add such comments in the proof of this theorem.

- There seems to be a typo in the proof — it should be $\mathbb{R}^{n_f}$ instead of $\mathbb{R}^n_f$.

2. Can the authors explain why they obtain lower metrics for GNN baselines than those presented in [2], if they use the same hyperparameter search space?

[2] Classic GNNs are Strong Baselines: Reassessing GNNs for Node Classification, NeurIPS 2024

---

> ### Author Response · Authors · 2025-11-20
> **Response to Reviewer 6Pr9**
>
> We thank the reviewer for highlighting the practicality of FAF as a simple yet strong baseline and for the insightful comments on theory and datasets. We address their concerns in the following points.
>
> ### Weaknesses
> 1. Related work extension
>
> We thank the reviewer for pointing out this recent benchmark, which was accepted only days before our submission and was therefore not covered in the original manuscript. We also note that one of the main goals of our paper is to criticise and analyse a wide range of existing datasets that much of the contemporary work still uses for evaluation; therefore, our choice of benchmarks is central to the paper, and it is not in our scope to criticise new attempts to provide better benchmarks. Our results are precisely supportive of continuing this line of work, as we mention in Section 4.3, and we are happy to incorporate it into the revision’s related work section as part of a new paragraph on improved benchmarking.
>
> 2. Experiment extension
>
> Due to time constraints in the rebuttal process, we are only able to provide 4 of the RL classification benchmarks as an illustration of the transferability of our results to other benchmarks. Moreover, we cannot provide the full hyperparameter sweep for our method (we perform some manual sweeps instead), nor all versions (we only evaluate mean+std and mean), so there might be even better FAFs than what we report here. Baselines are directly taken from the paper. While we cannot yet achieve the performance of GATs, we approach that of GCNs, and we improve upon their MLPs and NFA.
>
> Table 1: Test accuracy of 4 datasets from GraphLand benchmark (averaged over 10 runs).
>
> |                  | artnet-exp   | hm-categories | tolokers-2   | pokec-regions |
> |------------------|-------------:|--------------:|-------------:|--------------:|
> | ResMLP           | 35.07 ± 2.34 | 37.72 ± 0.18  | 41.16 ± 1.13 |  4.88 ± 0.01  |
> | ResMLP-NFA       | 38.25 ± 0.56 | 48.72 ± 0.38  | 48.14 ± 1.40 |  8.05 ± 0.03  |
> | GCN              | 43.09 ± 0.38 | 61.70 ± 0.35  | 51.32 ± 0.96 | 34.96 ± 0.38  |
> | GAT              | 46.62 ± 0.32 | 67.96 ± 0.33  | 53.78 ± 1.34 | 46.17 ± 0.32  |
> | FAF mean+std     | 41.56 ± 0.26 | 59.50 ± 0.15  | 52.74 ± 0.53 | 28.44 ± 0.22  |
> | FAF mean         | 39.25 ± 0.38 | 54.50 ± 0.13  | 50.72 ± 0.38 | 31.23 ± 0.18  |
>
>
>
> Table 2: Hyperparameters for FAF.
>
> |               | dropout | lr     | bn | hidden_channels | weight_decay | local_layers | mlp_layers |
> |---------------|---------|--------|----|-----------------|--------------|--------------|------------|
> | artnet-exp    | 0.7     | 0.01   | 1  | 256             | 0.01         | 3            | 2          |
> | hm-categories | 0.5     | 0.001  | 1  | 512             | 0.0005       | 3            | 3          |
> | tolokers-2    | 0.3     | 0.0001 | 1  | 512             | 5e-05        | 3            | 5          |
> | pokec-regions | 0 | 0.001 | 1 | 512 | 0.001 | 3 | 3 |
>
> ### Questions
>
> 1. We thank the reviewer for the helpful comments on the formulation of Theorem 1, which we have adapted accordingly in the revised manuscript.
>
> a. Concretely, we have replaced the term “unique” by a statement about the injectivity of $f$ and explained its meaning in the text below the theorem. The reviewer interpreted our statements correctly.
>
> b. The proof constructs the decomposition of $f$ now explicitly. We have also added an intuitive explanation in the main text.
>
> c. The typo is corrected.
>
> 2. Different metrics than the benchmark.
>
> For the baseline GNN methods, we do not perform a full hyperparameter sweep. Instead, we use the best configurations reported in the original work, as provided in their codebase. The baselines are run directly using the authors’ released implementations from their official GitHub repository (LUOyk1999/tunedGNN). Moreover, note that, because our study focuses on model selection, we have reported validation performance in the main text. To verify that we generalise equally well, we report train, validation, and test learning curves for our main method FAF4 and the GCN baseline in Appendix D Fig.6, confirming that validation and test metrics match throughout training (achieving higher test than validation accuracy in many cases). We are, of course, happy to also include the corresponding test-accuracy table in D.1, that (in line with Fig. 6) supports also all our insights: On all datasets, FAFs achieve performances within ±1% of the best classic GNN, except for those already mentioned in the main text (Citeseer, Roman-Empire, and Minesweeper), and Cora, which shows higher test than validation accuracy (see Fig. 6.h).

---

> > ### Comment · Reviewer_6Pr9 · 2025-11-24
> >
> > I thank the authors for extending the discussion of related work and for additional experiments on GraphLand benchmark. I hope that new results can find place in the final revision of the paper. My concerns regarding the predictive performance of GNNs taken from [1] have also been addressed. I believe that now graph ML community needs to improve the culture of empirical research by adopting proper tuning of standard GNNs and exploring more simple baselines that are relevant in practice. From my perspective, this study further develops the recent line of works on revisiting classic GNNs [1, 2] and makes valuable contribution by introducing a very simple approach based on graph aggregated features that performs on par with properly tuned GNNs. This exactly meets the current needs in graph ML research and can be very useful for the community. I increase my score accordingly.
> >
> > [1] Classic GNNs are Strong Baselines: Reassessing GNNs for Node Classification, NeurIPS 2024
> >
> > [2] Can Classic GNNs Be Strong Baselines for Graph-level Tasks? Simple Architectures Meet Excellence, ICML 2025

---

> > > ### Author Response · Authors · 2025-11-25
> > >
> > > We sincerely thank you for your encouraging assessment and very helpful feedback. We have added the GraphLand results in the revision as suggested (appendix E).

---

### Official Review · Reviewer_Vmh1 · 2025-10-27

**Soundness:** 3
**Presentation:** 4
**Contribution:** 3
**Rating:** 8
**Confidence:** 3

**Summary:**

This paper challenges the common view that neighborhood aggregation of GNNs must be learned, and introduces Fixed Aggregation Features (FAFs). FAFs are essentially non-trained feature aggregators (e.g. sum, mean) applied at fixed hops and transformed into tables.A Kolmogorov–Arnold analysis justifies this design, which appears to work empirically in many node classification datasets.

**Strengths:**

- challenges the established view of needing learned aggregations in GNNs with convincing arguments; in this respect, it can be considered innovative.
- the tabular representation is interpretable with standard tools (e.g. SHAP), which clearly adds value.
- the experiments are adequate in both in width (14 datasets) and depth (large set of hyperparameters), the design is fair and reproducible
- some insights (e.g. GNNs may overfit later aggregations) are definitely thought-provoking and might help designing better GNNs (or improve their benchmarks).
- the paper is clearly written and easy to follow.

**Weaknesses:**

- The way it is presented, the main finding of the paper appears to be confined to (transductive) node classification. A more broader characterization (e.g. an extension to graph classification or to inductive node-classification) would increase the significance of this work.

- The work (and especially its recommendation on using FAF baselines and reassess benchmarks) is connected with previous work on properly benchmarking GNNs for graph classification. In particular, [1] proposes simple baselines and similarly argues that current graph datasets are often inadequate. These connections should be acknowledged in the Related Works section.

[1] Errica et al. A Fair Comparison of Graph Neural Networks for Graph Classification. ICLR 2020

**Questions:**

No questions, besides the relatively minor weaknesses detailed above. I believe this paper is a very solid contribution to the field already as-is.

---

> ### Author Response · Authors · 2025-11-20
> **Response to Reviewer Vmh1**
>
> We thank the reviewer for their very positive assessment and for recognising the innovation in questioning the need for learned aggregations, as well as the clarity and breadth of our experiments. We address their concerns in the following points.
>
> ### Weaknesses
> 1. Graph classification and inductive settings
>
> Our theory applies to any task that learns multiset functions over neighborhoods. In our experiments, we focus on node classification for two main reasons. First, these are the benchmarks on which GNNs have been shown to be competitive with more complex architectures in Luo et al. (2024), so they are amenable to simple models for which we have strong, well-tuned hyperparameters. Second, node-classification datasets typically provide rich features that depend on neighborhood distributions. Thus, non-injective but commonly used reducers such as mean and sum still convey highly informative distributional signals. Regarding inductive settings, they would require computing the new aggregation rounds at test time. We would not have access to the test node features when precomputing training aggregations. Otherwise, our approach is just as feasible as in the transductive case. We are happy to include this discussion in the revised Appendix B.3.
>
> 2. Related work extension
>
> We thank the reviewer for pointing out this work, which we now include in our related work in the revised manuscript as part of a new paragraph on improving benchmarks:
>
> Our work also connects to the growing literature on properly benchmarking GNNs and what constitutes a meaningful graph-learning dataset. For graph classification, \citet{Errica2020A} show that, under controlled protocols, simple and even structure-agnostic baselines can rival complex GNNs, suggesting that common benchmarks often fail to exploit graph structure. Recent analyses likewise warn that graph learning risks losing relevance without application-grounded benchmarks \citep{bechler-speicher2025position} and principled criteria for dataset quality beyond accuracy \citep{coupette2025no}. On the dataset side, \citet{bazhenov2025graphland} recently introduced industrial node property prediction benchmarks with graph-agnostic baselines. Their neighborhood feature aggregation (NFA), which augments tabular models with one-hop aggregated neighbor statistics, can be seen as a one-hop instance of our FAF construction. Concurrently, \citet{anonymous2025graphbench} propose a suite of benchmarks to standardize graph-learning evaluation. In this context, our FAF approach serves as a simple stress test of whether proposed graph benchmarks genuinely benefit from learned message passing, and we argue that such well-tuned, fixed, multi-hop baselines should be routinely included when assessing new graph models and datasets.

---

> > ### Comment · Reviewer_Vmh1 · 2025-11-21
> > **Thanks**
> >
> > Thank you for the additional clarifications, I have no remaining concerns and I believe this submission should be accepted. Good luck!

---

> > > ### Author Response · Authors · 2025-11-25
> > >
> > > We sincerely thank you for your encouraging assessment and very helpful feedback.

---

### Author Response · Authors · 2025-11-20
**Common Response to All Reviewers**

We thank all reviewers for their careful and constructive feedback.

There is broad agreement that our work (i) introduces a simple but strong class of fixed-aggregation baselines (FAFs) (6Pr9, fmA9, 78XL), (ii) provides an extensive and fair empirical comparison against well-tuned GNNs (Vmh1, 6Pr9, fmA9), (iii) raises important questions about what current graph benchmarks truly measure (6Pr9, fmA9), (iv) unveils a novel view of graph aggregation in terms of Kolmogorov–Arnold representations (78XL), and (v) challenges requirements for injectivity while exposing other empirical obstacles (Vmh1, 6Pr9).

In response to the main concerns, we (a) clarify the scope and connect more explicitly to recent benchmarking work, (b) tighten the link between our Kolmogorov–Arnold-based theory and the empirical findings, extending our Section 4.3, (c) provide new experiments on four GraphLand benchmarks showing that FAFs remain close to GCNs and clearly improve over MLP and NFA baselines, (d) discuss efficiency with added runtime results, and (e) clarify our experimental protocol (validation vs. test, reuse of best configs from Luo et al. (2024) and the fact that only FAF4 is tuned, with other FAFs treated as ablations). Overall, we believe the revisions strengthen the case for FAFs as standard, interpretable tabular baselines that stress-test whether graph benchmarks genuinely benefit from learned message passing, and provide a novel angle on graph aggregation.

---

### Author Response · Authors · 2025-12-03

Dear Area Chair, Senior Area Chair, and Program Chairs,

Thank you for your time and effort in reviewing our paper and the rebuttal. We would like to summarize our work's main contribution and the discussion of the review process.

**Brief summary of contribution:** This submission introduces Fixed Aggregation Features (FAFs), a simple, training-free neighborhood aggregation scheme that converts node classification tasks into tabular learning problems. A downstream MLP is then trained on these fixed multi-hop aggregated features. Despite its simplicity, the method achieves performance comparable to well-tuned GNNs across most standard node-classification datasets, which has several relevant implications for the research community. To analyze why this approach can be sufficient, we provide a theoretical perspective based on Kolmogorov–Arnold representations, arguing that expressive, lossless fixed aggregation schemes exist in principle, and discuss why simple non-injective reducers (e.g., mean aggregation) suffice in practice on current benchmarks.

**Strengths across reviews:** Three reviewers (Vmh1, 6Pr9, fmA9) expressed **strongly positive assessments** following the rebuttal and either maintained (Vmh1) or raised (6Pr9, fmA9) their scores to **8**. In particular, reviewers 6Pr9 and fmA9 were *“impressed”* by the new GraphLand results added during the rebuttal, which show that FAFs transfer reasonably well to new datasets and outperform the MLP and NFA baselines. We also addressed their earlier concerns, for instance, by clarifying the experimental setup and by *“increasing their appreciation”* of the ties between the theoretical analysis and the empirical results. We would like to emphasize that all three reviewers highlight the **usefulness** of our work for the graph learning community. They view the paper as a **valuable and timely** contribution that meaningfully advances the ongoing push for stronger baselines and more principled benchmarking.

**Reviewer 78XL:** Reviewer 78XL increased their score from a 2 to a **4** during the rebuttal phase. Importantly, they stated that they **will not oppose acceptance**, and updated portions of their assessment after the discussion period, including clarifications on the introduction and the experimental setup. Their remaining reservations relate primarily to stylistic preferences regarding the paper’s narrative. For instance, they suggest that the work could be split into two papers, whereas we argue that our theoretical and empirical insights are tightly interdependent and that only by considering both together can one fully understand the scope of our conclusions. The reviewer is more interested in a theory-centric exploration of injective fixed functions than in our combined message that simple, non-injective FAFs already match tuned GNNs on current benchmarks. We view such an exploration as a natural next step that builds on our work. However, progress on better aggregation schemes is currently likely limited by the lack of more complex benchmarks to improve upon, as we have shown. These are viable paths for future research directly inspired by our insights. Moreover, the reviewer expressed a preference for an additional baseline (specifically GESN), which we ultimately provided during the discussion and in the revision before the discussion freeze. We find that our proposed FAF features consistently outperform GESN, suggesting that FAFs have a strong and useful inductive bias for typical node classification benchmarks. Unfortunately, a subsequent discussion on this was cut short, so we could not further engage and ensure their satisfaction with it.

**Conclusion:** We believe we have addressed all raised concerns and are grateful for the reviewers’ feedback that improved the empirical evidence and the clarity of our message. Our contributions are relevant to the ICLR community, as our work calls for further benchmarks that can move the graph-based learning field forward through concrete insights for their design, and we advocate for simple, non-trainable features (FAFs) as strong baselines for measuring future progress.

---

### Meta-Review · Area_Chair_qXoQ · 2025-12-21

**Summary:**

The paper proposes Fixed Aggregation Features (FAFs): precompute multi-hop fixed neighborhood stats (often mean/max/std) and train an MLP; matches or beats well-tuned GNNs on many node-classification benchmarks and argues current benchmarks often don’t need learned message passing.

Strengths
* Very simple, strong baseline; clear “tabularize graphs + MLP” takeaway.
* Broad, mostly careful empirical study; rebuttal adds GraphLand and runtime evidence.
* Valuable benchmark critique: many datasets solvable with shallow, non-trainable aggregations.

Weaknesses
* Main dissent: theory–practice disconnect (injective KA theory vs non-injective reducers driving results); narrative/positioning unclear.
* Interpretability claims not convincing (incl. initial minesweeper feature misunderstanding).
* Baseline completeness/eval presentation confusion (GESN comparison pressure; validation-heavy reporting initially).

Despite strong empirical signal and several high scores after rebuttal, persistent concern about contribution clarity and coherence (plus interpretability/baseline presentation issues) prevents consensus.

**Reviewer Scores:**

n/a

---

### Decision · Program_Chairs · 2026-01-26

Reject